# A recombined *Sr26* and *Sr61* disease resistance gene stack in wheat encodes unrelated *NLR* genes

Jianping Zhang [1,2], Timothy C. Hewitt [1,2], Willem H. P. Boshoff[3], Ian Dundas[4], Narayana Upadhyaya [2], Jianbo Li[1], Mehran Patpour [5], Sutha Chandramohan[2], Zacharias A. Pretorius [3], Mogens Hovmøller[5], Wendelin Schnippenkoetter [2], Robert F. Park [1], Rohit Mago [2], Sambasivam Periyannan[2], Dhara Bhatt[2], Sami Hoxha[1], Soma Chakraborty[2], Ming Luo[2], Peter Dodds [2], Burkhard Steuernagel [6], Brande B. H. Wulff[6], Michael Ayliffe[2], Robert A. McIntosh[1], Peng Zhang [1✉] & Evans S. Lagudah [1,2✉]

The re-emergence of stem rust on wheat in Europe and Africa is reinforcing the ongoing need for durable resistance gene deployment. Here, we isolate from wheat, *Sr26* and *Sr61*, with both genes independently introduced as alien chromosome introgressions from tall wheat grass (*Thinopyrum ponticum*). Mutational genomics and targeted exome capture identify *Sr26* and *Sr61* as separate single genes that encode unrelated (34.8%) nucleotide binding site leucine rich repeat proteins. *Sr26* and *Sr61* are each validated by transgenic complementation using endogenous and/or heterologous promoter sequences. *Sr61* orthologs are absent from current *Thinopyrum elongatum* and wheat pan genome sequences, contrasting with *Sr26* where homologues are present. Using gene-specific markers, we validate the presence of both genes on a single recombinant alien segment developed in wheat. The co-location of these genes on a small non-recombinogenic segment simplifies their deployment as a gene stack and potentially enhances their resistance durability.

[1] Plant Breeding Institute, School of Life and Environmental Sciences, University of Sydney, Cobbitty, NSW, Australia. [2] CSIRO Agriculture & Food, Canberra, ACT, Australia. [3] Department of Plant Sciences, University of the Free State, Bloemfontein, South Africa. [4] School of Agriculture, Food and Wine, University of Adelaide, Urrbrae, SA, Australia. [5] Department of Agroecology, Aarhus University, Slagelse, Denmark. [6] John Innes Centre, Norwich Research Park, Norwich, UK. ✉email: peng.zhang@sydney.edu.au; evans.lagudah@csiro.au

The emergence of widely virulent *Puccinia graminis* f. sp. *tritici* (*Pgt*) races over the past two decades[1,2] has motivated global efforts to identify effective stem rust (Sr) resistance genes. During the last seven years, nine seedling (or all stage) Sr genes (*viz. Sr13, Sr21, Sr22, Sr33, Sr35, Sr45, Sr46, Sr50,* and *Sr60*) have been cloned, eight of which encode nucleotide-binding, leucine-rich-repeat (NLR) immune receptors[3–9]. *Sr60* is an exception that encodes a tandem kinase protein[10]. These genes were targeted due to their effectiveness against Ug99 and other *Pgt* races. Their sequences now provide perfect markers and diagnostic tools for marker-assisted breeding. However, the subsequent appearance of new, diverse virulent isolates means that most of these cloned *Sr* genes have been overcome by at least one *Pgt* race within and/or beyond the Ug99 lineage. Consequently, there is an ongoing need to expand resistance resources and to enhance gene stewardship through co-deployment of multiple resitance (R) genes, rather than single genes, to increase resistance durability.

Grass species related to wheat carry sources of resistance that can be transferred to wheat. *Sr26* is derived from tall wheat grass [*Thinopyrum ponticum* ($2n = 10× = 70$)], and its introgression into common wheat as a T6AS.6AL-6Ae#1 translocation chromosome is an early example of a transfer of resistance from a wheat wild relative[11,12]. *Sr26* was transferred to wheat chromosome 6A by seed irradiation in the early 1960s and this resistance has remained effective against all tested *Pgt* races, including those in the Ug99 group[11,13–15]. A second *Th. ponticum*-derived Sr gene, *Sr61* (previously designated *SrB*), was identified in South African wheat accession W3757, which carries a 6Ae#3 (6D) chromosome substitution[16]. No *Pgt* virulence has been reported for *Sr61* either. *Sr26* has been deployed in a number of Australian wheat cultivars since 1971 and has likely fulfilled the definition of durable resistance[17]. *Sr61* on the other hand has not been deployed in a cultivar. Since genetic dissection of genes within alien segments in wheat was not possible due to lack of recombination, the question arose as to whether the apparent durability of such resistances might be due to multiple genes rather than a single gene. Resistance in W3757 was located on chromosome 6Ae#3, making it possible that *Sr26* and *Sr61* are alleles or related paralogues[18].

It is noteworthy that neither of these *Th. ponticum* segments can recombine with wheat chromatin; however, they can recombine with each other when both are present in wheat. Recently, a recombinant line was developed in which *Sr61* was transferred from 6Ae#3 to a T6AS.6AL-6Ae#1 translocation segment by homologous recombination[19] with the 6Ae#1 segment carrying *Sr26*. However, as there is no current *Pgt* race known to be virulent to either resistance gene, it was not possible to unambiguously determine whether the recombinant introgression carried a single or both genes. Molecular markers developed for *Sr26* and *Sr61* were not reliable indicators of the Sr genes per se, as they were based on DNA sequences that were located at unknown positions on the entire non-recombinogenic alien segments or alternatively, present in large linkage blocks of unordered markers that cosegregated with the resistance[19–22]. The ambiguity surrounding the presence of both genes in the potential recombinant segment was therefore difficult to resolve using traditional methods. After more than 40 years following their characterization in the 1980s we have used cloning to determine the relationship existing between these two very useful R genes.

Here, we show that *Sr26* and *Sr61* encode unrelated *NLR* genes that have been combined in a single *Th. ponticum* segment representing a natural gene stack, that simplifies codeployment of these two resistance genes and potentially enhances their resistance durability.

## Results

**Identification of a *Sr26* candidate gene by MutRenSeq.** Conventional map-based cloning of *Sr26* and *Sr61* in wheat derivatives was not feasible due to the absence of recombination between common wheat and alien chromosome segments. Target-sequence Enrichment and Sequencing (TEnSeq) pipelines developed in recent years have overcome such constraints to map-based cloning[23]. Here we used mutational genomics and targeted exome capture of NLR immune receptor genes, a method termed MutRenSeq[8], for isolation of *Sr26* and *Sr61*. For *Sr26* cloning we screened 1270 $M_2$ lines and identified five susceptible ethyl methanesulfonate (EMS)-induced mutants from the *Sr26*-carrying wheat genetic stock, Avocet+Lr46[22] (Fig. 1a and Supplementary Table 1). One mutant (150S) carried a deletion of linked marker #43[20]. Because there was no information regarding the genetic distance from the linked marker, we hypothesized that simultaneous loss of a marker with loss of the resistance was indicative of a deletion of unknown size[20]. The five mutants, together with wild-type Avocet+Lr46, were subjected to NLR gene exome capture, sequencing and alignment using MutantHunter[8] (Supplementary Fig. 1a). A single contig of 2466 bp that was absent from the putative deletion mutant 150S contained a single nonsynonymous nucleotide change in each of the remaining four mutants. The full length gene isolated from Avocet+Lr46 encodes a 935 amino acid (aa) protein consisting of a coiled-coil (CC) domain at the N-terminus, a NB-ARC domain and LRR motifs at the C-terminus (CNL) (Fig. 1b). Three mutants encoded amino acid changes in conserved motifs in the NB-ARC domain; Ala311Thr (RNBS-C motif) in mutant 128S and a common Ser431Asn (RNBS-D motif) mutation in mutants 70S and 12S, suggesting the latter two mutants were sibs that inherited the same mutation event. The nucleotide change in mutant 499S was a premature stop codon (Supplementary Fig. 2).

**Transgenic validation of the *Sr26* candidate gene.** A transgenic complementation experiment was undertaken to confirm the function of the *Sr26* gene candidate in wheat cultivar Fielder. The assembled genomic sequence available for the *Sr26* candidate included 917 bp 5′ of the translational start codon and 263 bp 3′ of the stop codon, and therefore may lack sufficient regulatory sequences for appropriate gene expression (Fig. 2a). To ensure candidate gene expression, three constructs were used to produce transgenic plants (Fig. 2a). One construct comprising the available native sequences described above was designated *Sr26:NativeRE* (*Regulatory Elements*). The other two, designated *Sr26:Sr22RE* and *Sr26:Sr33RE*, combined the *Sr26:NativeRE* construct with 5′ and 3′ regulatory elements derived from wheat stem rust resistance genes *Sr22* and *Sr33*, respectively[7,8] (Fig. 2a). A previous report had shown that *Sr45* gene function was retained when expressed using *Sr33* regulatory elements[24], suggesting these heterologous regulatory sequences could be used for *Sr26* expression.

We generated 21, 22, and 14 independent primary transgenic Fielder lines carrying the *Sr26:NativeRE*, *Sr26:Sr22RE*, and *Sr26:Sr33RE* constructs, respectively. All 57 independent $T_0$ plants were resistant to a *Pgt* isolate of race 98-1,2,3,5,6+Sr50, whereas all non-transformed Fielder controls were susceptible (Fig. 2b and Supplementary Table 2). These data indicated that the minimal endogenous promoter and terminator sequences used for the *Sr26:NativeRE* construct were sufficient for functional *Sr26* expression, as was the addition of *Sr22* and *Sr33* promoter and terminator sequences. To confirm the resistance cosegregated with the transgene, two $T_1$ families of 15 and 16 progeny from line PC235-5 and PC235-4 containing the *Sr26:NativeRE* transgene were analysed (Supplementary Fig. 3a, b). Transgene

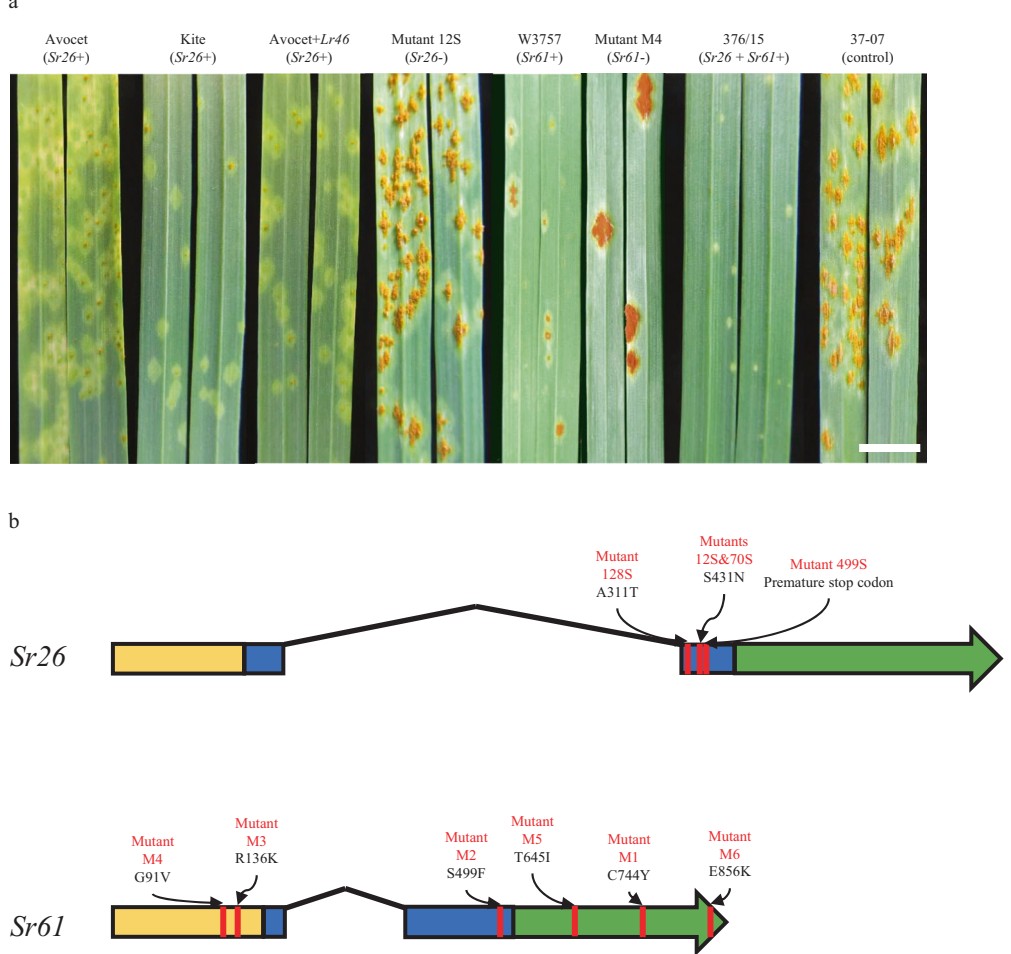

**Fig. 1 Rust response phenotypes conferred by the *Sr26* and *Sr61* parental sources and mutant derivatives, and gene structures of *Sr26* and *Sr61* candidate genes. a** Abaxial seedling leaf surfaces of wild-type and representative EMS-generated susceptible mutants for *Sr26* and *Sr61*, together with recombinant 376/15 inoculated with *Pgt* race PTKST. Avocet, Kite, Avocet+Lr46, W3757, and recombinant 376/15 showed low infection types (small pustules or flecking), whereas *Sr26* mutant 12S, *Sr61* mutant M4 and susceptible control line 37-07 all showed high infection types (large pustules). Bar shows 1 cm. **b** Gene structures of candidate *Sr26* and *Sr61* genes. Exons are shown as solid boxes with the terminal exon indicated as an arrow to show the direction of transcription whereas introns are shown as solid black angled lines. The position of mutations in mutant lines are shown in red with predicted amino acid substitutions caused by nonsense mutations and premature stop codon shown above. CC domain, NB domain, and LRR motifs are shaded upon the exons with yellow, blue, and green colors.

copy number was determined in PC235-5 by quantitative PCR and a minimum of two transgene loci, that segregated independently were detected based upon gene copy number (Supplementary Fig. 3c). All transgenics with at least one transgene copy (i.e., hemizygous) were resistant whereas the null segregant and Fielder control were susceptible (Supplementary Fig. 3c). Forty-six $T_1$ plants derived from four independent transgenic events, two *Sr26:Sr22RE* and two *Sr26:Sr33RE* events, were all resistant to *Pgt* race 98-1,2,3,5,6+Sr50 whereas all Fielder controls lacking the transgene were susceptible (Supplementary Fig. 3d, e). These latter heterologous regulatory sequences caused no change in the *Sr26* resistance phenotype seen in transgenic plants (Supplementary Fig. 3a, b, d, e).

**Identification of an *Sr61* candidate gene by MutRenSeq.** We identified eight susceptible EMS mutants derived from line W3757 among 1837 $M_2$ lines screened. Two mutants contained deletions of a previously reported marker, MWG798, linked to *Sr61*[19]. The remaining six mutants (M1–M6) were potential point mutations and together with wild-type W3757 were analysed by NLR gene capture and sequencing (Fig. 1a and Supplementary

Fig. 1b). A single contig was identified that contained nucleotide changes in five of the six mutants (M1–M5) (Supplementary Fig. 1b). The full length gene of 3519 bp isolated from W3757 encodes an 880 aa protein containing a coiled-coil (CC) domain, NB-ARC domain and LRR motifs (Fig. 1b). A nucleotide substitution was subsequently identified in the sixth mutant (M6) in the sequence present in the full-length gene that was absent in the original MutRenSeq contig. Each mutant contained non-synonymous nucleotide changes that caused amino acid substitutions in either the CC (M3, M4), NB-ARC (M2), or LRR (M1, M5, and M6) domains (Fig. 1b and Supplementary Fig. 2).

**Transgenic validation of the *Sr61* candidate gene.** Transgenic complementation in wheat cultivar Fielder was used to confirm the function of the *Sr61* gene candidate. The assembled genomic sequence available for the *Sr61* candidate encoded only 354 bp 5′ and 67 bp 3′ of the gene translation start and stop codons, respectively. To ensure the expression of the candidate gene a heterologous construct designated *Sr61:Sr26RE* was produced encoding the *Sr61* genomic sequence, and the 5′ and 3′ *Sr26* regulatory elements shown above to function in the *Sr26:NativeRE*

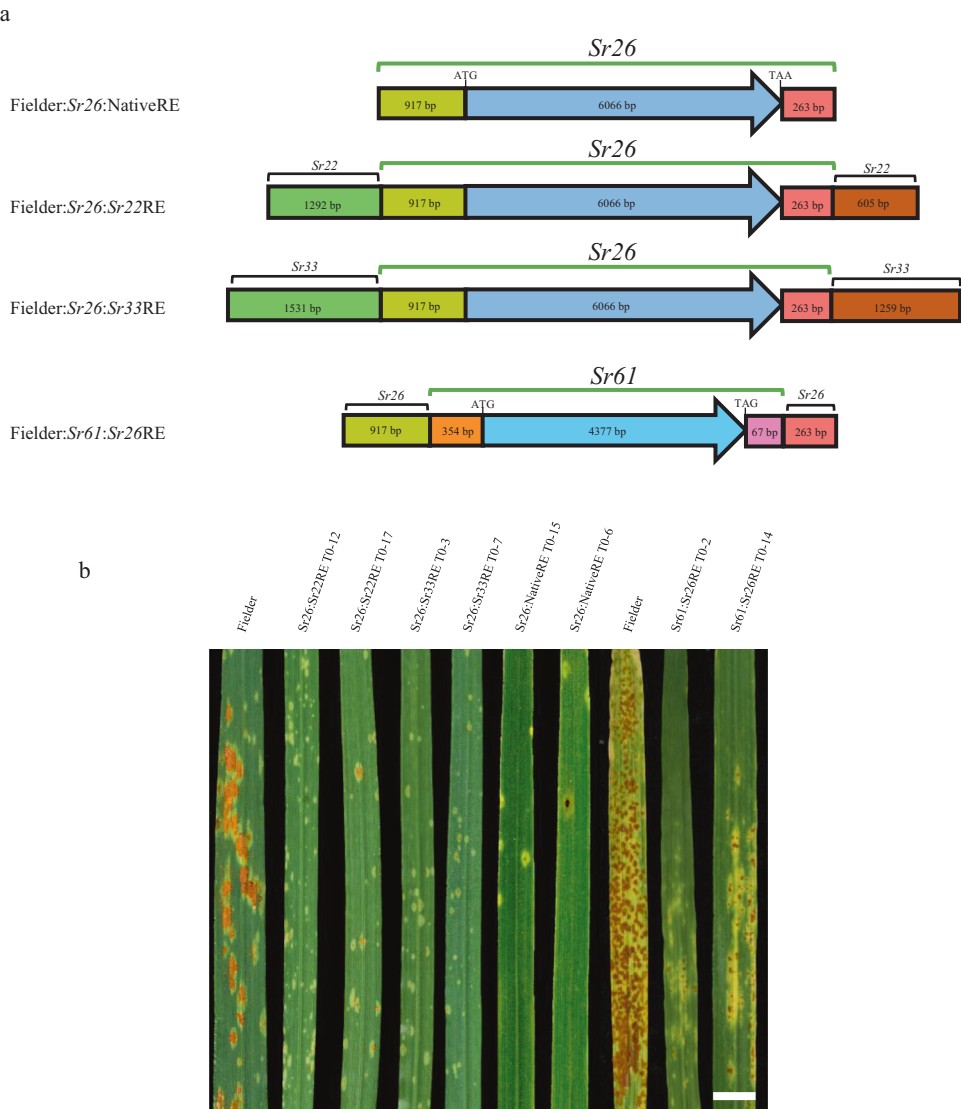

**Fig. 2 Validation of the *Sr26* and *Sr61* candidate genes by transformation. a** Four constructs used for wheat transformation. The three *Sr26* constructs encoded the *Sr26* gene candidate and either its native regulatory sequences (*Sr26:NativeRE*), regulatory sequences from *Sr22* (*Sr26:Sr22RE*) or regulatory sequences from *Sr33* (*Sr26:Sr33RE*). A single construct was used for the *Sr61* gene candidate under the regulatory control of *Sr26* 5′ and 3′ regulatory elements (REs). 5′ REs and 3′ REs are indicated by black brackets with sizes in bp indicated for all four constructs. *Sr26* and *Sr61* intron/exon coding regions are shown as light and dark blue arrows. **b** Phenotypic responses of representative $T_0$ plants produced for all four constructs and inoculated with *Pgt* race 98-1,2,3,5,6+Sr50 along with non-transgenic Fielder control lines. All lines except the susceptible Fielder control showed low infection types. Photographed 12 days of post-inoculation. Bar shows 1 cm.

construct (Fig. 2a). We generated 21 independent primary transgenic lines carrying the *Sr61:Sr26RE* construct. From these 21 lines, 14 $T_0$ plants at a similar growth stage were selected and inoculated with the *Pgt* race 98-1,2,3,5,6+Sr50 isolate. All 14 lines were resistant whereas non-transformed Fielder controls were susceptible (Fig. 2b and Supplementary Table 2). Ten transgenic plants from three independent $T_1$ families were chosen to compare *Pgt* resistance phenotypes with transgene copy number and expression (Supplementary Fig. 3f). Eight plants contained at least one transgene copy and all were resistant except two plants, Sr61:Sr26RE $T_1$-6-7 and Sr61:Sr26RE $T_1$-17-11, which had a high infection type. qRT-PCR was then performed on all 10 $T_1$ lines and the results revealed that plant Sr61:Sr26RE $T_1$-17-11 had a single transgene copy that had very low expression, when compared with other transgene positive plants (Supplementary Fig. 3f). A resistant sib of this plant ($T_1$-17-2) contained two additional transgene copies suggesting that at least two independent transgene loci

segregated in this family, one of which had sufficient expression for *Sr61* resistance (Supplementary Fig. 3f). Similarly, plant Sr61:Sr26RE $T_1$-6-7 showed no detectable *Sr61* transcript expression, despite an apparently high transgene copy number, whereas its sibling line Sr61:Sr26RE $T_1$-6-3 expressed the transgene normally, again suggesting the segregation of independent expressing and non-expressing transgenic loci in this line (Supplementary Fig. 3f).

**Application of *Sr26* and *Sr61* in developing wheat breeding germplasm.** To facilitate the use of *Sr26* and *Sr61* in breeding and permit their reliable identification in the recombinant introgression segment described previously[19], we developed gene-specific markers for each gene. For *Sr26*, a 1580 bp PCR amplicon was identified that flanked the intron I–exon II junction and was highly specific for *Sr26* (Supplementary Fig. 4a). For *Sr61*, a marker with an amplicon size of 278 bp located in the first exon was confirmed to be *Sr61*-specific (Supplementary Fig. 4b and

Supplementary Table 3). Using these markers, we confirmed the presence of both *Sr26* and *Sr61* in the recombinant line (Supplementary Fig. 4). Molecular cytogenetic analysis showed that the alien segment in the recombinant line was substantially smaller than that in both the T6AS.6AL-6Ae#1 translocation[22] and 6Ae#3 chromosome substitution lines (Supplementary Fig. 5).

**Sr26 and Sr61 homologs in the Thinopyrum and Chinese Spring genomes**. We searched for homologs of *Sr26* and *Sr61* in the recently available *Th. elongatum* genome sequence[25] and the Chinese Spring genome. Sequences with greatest homology to *Sr26* coding sequences was located on the distal end of the group 6 chromosome of the E subgenome, and on the distal end of chromosome 6B in Chinese Spring. This is consistent with high synteny existing between the group 6 chromosomes of wheat (6A, 6B, and 6D) and chromosome 6E of *Th. elongatum* (Supplementary Table 4), and the location of *Sr26* in the T6AS.6AL-6Ae#1 translocation. However, the best hits of *Sr61* were located on the proximal region of 6E in *Th. elongatum* or on 2D or 7D in Chinese Spring and showed poor matches with less than 61 and 69% of the gene covered. This suggests that these genomes do not include orthologous genes to *Sr61*.

**Phylogenetic analysis of Sr26 and Sr61**. To determine the evolutionary relationship of Sr26 and Sr61 to other plant known R proteins of the CNL class, we generated a phylogenetic tree based on the alignment of 123 proteins and used the L6 flax rust resistance Toll/interleukin-1 receptor (TIR) NLR protein as an outgroup (Fig. 3a and Supplementary Data 1)[26]. Three large clades (Clade I–III) and a small clade (Clade IV) were identified (Fig. 3a). Although both *Sr26* and *Sr61* originated from tall wheat grass, the most closely related R protein to Sr26 was encoded by the *Triticum turgidum* ssp. *dicoccum* Sr13 gene (58.46% identity at protein level) (Fig. 3b). The Sr61 protein is less similar to both

Sr13 (35.21%) and Sr26 (34.81%), but all three proteins are members of a clade that includes Sr22, Sr33, Sr35, Sr50, Sr46, and proteins encoded by the barley *Mla R* gene family (Fig. 3a, Clade I). By contrast Sr21 and Sr45 are in a clade more closely related to wheat powdery mildew R proteins encoded by *Pm3* alleles (Fig. 3a, Clade II).

**Resistance spectra of Sr26 and Sr61**. The resistance responses conferred by *Sr26* and *Sr61* were tested against six and eight *Pgt* isolates, respectively, including races PTKST (collected in South Africa), TTRTF (collected in Italy and Eritrea), and TTKTT (collected in Kenya) (Supplementary Fig. 6 and Supplementary Tables 5 and 6). Screening of wild-type and mutant lines demonstrated that *Sr26* unequivocally provided resistance to all six isolates, whereas *Sr61* gave resistance to at least five of eight isolates. Additional resistance genes in the background of line W3757 were epistatic to *Sr61* masking its effectiveness against the other three isolates. In all assays, the three recombinant plants containing both gene candidates were consistently more resistant than lines with only one respective gene (Fig. 1 and Supplementary Figs. 6 and 7).

## Discussion

Among the many wild relatives of *T. aestivum*, perennial wheat grasses (polyploid *Thinopyrum* spp.) are a recognized source of desirable traits for tolerance of wheat to both biotic and abiotic stresses, such as diseases, pests, salinity, and drought[27]. Remarkably, six stem rust resistance genes (*Sr24*, *Sr25*, *Sr26*, *Sr43*, *Sr44*, and *Sr61*), four leaf rust resistance genes (*Lr19*, *Lr24*, *Lr29*, *Lr38*), two powdery mildew resistance genes (*Pm40*, *Pm43*), two barley yellow dwarf virus resistance genes (*Bdv2*, *Bdv3*), and a recently cloned wheat Fusarium head blight resistance gene (*Fhb7*) were transferred from *Th.* spp. to wheat as summarized by Wang et al.[25]. The recently published diploid *Th. elongatum* genome assembly will further facilitate the transfer of genetic

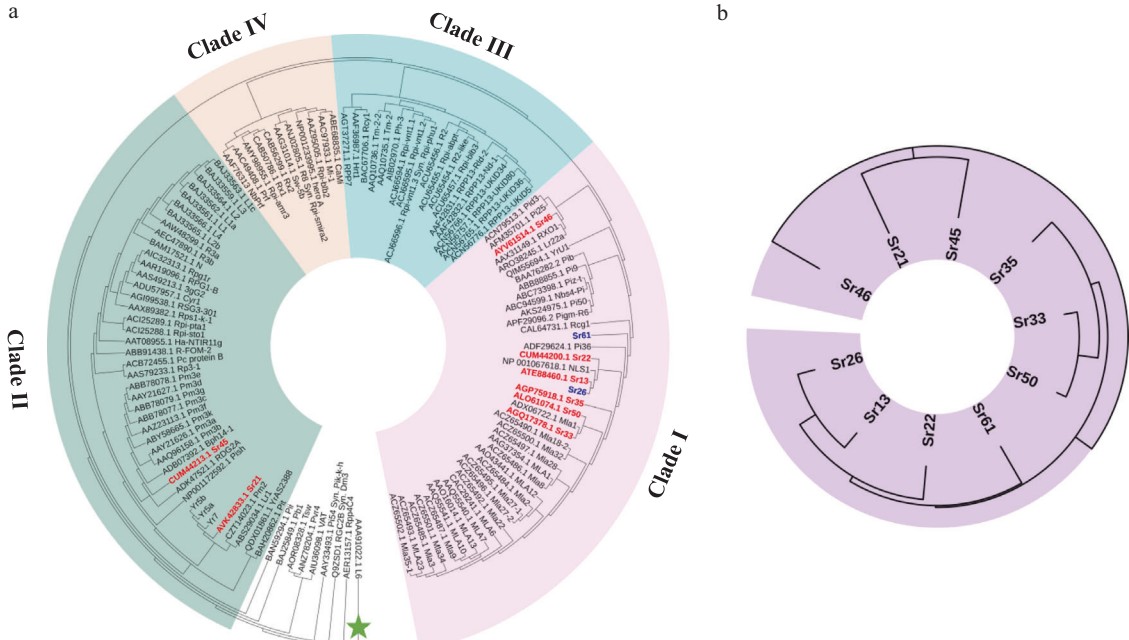

**Fig. 3 Phylogenetic relationship between Sr26, Sr61 and other CC-NLR immune receptors from plants. a** Comparison of 123 NLR type proteins formed three large clades (I–III) and one smaller clade (IV). The TIR type NLR protein encoded by the *Linum usitatissimum* L6 rust resistance gene was used as an outgroup (green star). Previously cloned wheat stem rust genes are in red script; *Sr26* and *Sr61* are in blue; and Clades I, II, III, IV are shaded in purple, green, blue, and light orange, respectively. **b** Simplified phylogenetic tree showing the evolutionary relationship between all ten cloned wheat stem rust *R* genes that encode CC-NLR proteins.

resources from *Thinopyrum* spp. into cultivated wheat[25]. Here, we report the characterization of two such *Th. ponticum* stem rust resistance genes, *Sr26* and *Sr61*, and show that they encode unrelated NLR proteins.

Using cytogenetic analysis we identified a close to entire chromosome arm 6Ae#1 present in Avocet+Lr46 and a complete chromosome 6Ae#3 in W3757 (Supplementary Fig. 5). These large chromosomal segments raised the possibility that the *Sr26* and *Sr61* resistance phenotypes could each be conferred by more than a single resistance gene carried within the large non-recombinogenic region. For example, *Sr32* resistance was first identifed as conferred by a large introgressed segment of *Aegilops speltoides* chromosome 2S#1 introduced by E.R. Sears[28]. Subsequent reduction of this segment by homoeologous recombination revealed that two different regions of this segment contain functional resistance genes, one of which was designated as *Sr32* and the second as *SrAest1t*, indicating that the orginal introgression carried at least two resistance genes[29]. Similarly, three different stem rust resistance genes, *Sr39, SrAest7t*, and *Sr47*, were also identifed in translocation involving a 2S chromosome (2S#2) derived from a different *Ae. speltoides* accession[29]. However, for each of the *Th. ponticum* segments, all mutants susceptible to *Pgt* race 34-1,2,3,4,5,6,7 contained deficiencies in a single NLR gene, confirming that a single gene in each segment confers resistance to this isolate. Screening of these mutants with geographically diverse *Pgt* races showed that all six *Pgt* races screened were virulent on the Sr26− Avocet+Lr46 mutants, suggesting that there is likely only one resistance gene on the original *Sr26* introgression segment. In contrast, 5 out of 8 *Pgt* races were virulent on W3757 mutants, indicating the presence a single gene effective against these isolates. However, additional resistance in W3757 masked the phenoype of *Sr61* in tests with three *Pgt* isolates. It is unknown if this additional resistance is also associated with the *Th. ponticum* introgression. Previously Singh and McIntosh postulated the presence of *Sr6* (located on chromosome 2D) and *Sr12* (chromosome 3B) in addition to *Sr61* in W3757[16]. The three *Pgt* isolates (races TTKTF, TTKTT, and TTRTF) that were avirulent to W3757 mutants (Supplementary Table 5) are each virulent to *Sr6* precluding this gene as providing background resistance whereas their pathogenicity to *Sr12* was uncharacterized[28]. Nevertheless, given its unusual requirment of lower tempeature for incubation period, *Sr12* also seems unlikely to provide seedling resistance in any of our disease resistance tests[28]. It remains possible that additional unknown resistance genes could be present in W3757 either on the introgressed segment or elsewhere in the genome. Any other resistances encoded on these introgressions provide limited resistance when compared with the two genes we have isolated, thus, we have designated these genes *Sr26* and *Sr61*, respectively.

Due to only short 5′ and 3′ noncoding sequences being available from both *Sr26* and particularly *Sr61*, we used heterologous regulatory sequences from *Sr22* and *Sr33* for transgene expression. Luo et al.[30] recently showed substantial variation existed between the expression levels of a number of different NLR encoding *Sr* genes, with the *Sr50* gene expressed at an approximately 10-fold higher level when compared with *Sr22*. Using *Sr26* regulatory elements we saw an approximately 40% reduction in *Sr61* transgene expression in resistant transgenics compared with the endogenous gene present in W3757; however, the resistance phenotype appeared unchanged. Similarly *Sr22* and *Sr33* regulatory sequences were sufficient to express *Sr26* and provided levels of resistance similar to the *Sr26* transgene containing native, albeit short, regulatory sequences and the endogenous *Sr26* gene in Avocet.

The tertiary wheat genepool genes, *Sr26* and *Sr61*, encode NLR proteins, like eight of the nine previously cloned wheat Sr genes[3–9]. Amongst cloned wheat stripe rust all-stage resistance genes, the *Yr15* gene encodes a tandem kinase domain protein

and *Yr5a*, *Yr5b*, and *Yr7* encode NLR proteins with integrated BED domain motifs[31,32]. In contrast, *YrU1* and *YrAS2388* encode canonical NLR proteins[33,34]. A number of NLR proteins are also the products of wheat leaf rust all-stage resistance genes (i.e., *Lr1*, *Lr21*) while *Lr10* requires dual NLR proteins for function[35–37]. Therefore, NLR proteins predominate in providing all-stage resistance to all three fungal rust species in wheat and its immediate ancestors. However, other proteins can also contribute, such as the recently reported membrane bound ankyrin repeat protein encoded by *Lr14a*[38].

Combining multiple *R* genes is a widely accepted gene stewardship strategy to enhance the durability of resistance on the basis that the simultaneous defeat of two or more effective *R* genes is less likely than defeat of either gene alone. One compelling reason for cloning *R* genes is the opportunity to combine them into transgenic cassettes to allow multiple *R* genes to be selected as a single "trait" in breeding[30]. The most desirable Sr genes to be combined into a transgene cassette are those with broad effectiveness against diverse *Pgt* races. Virulence to most of the currently cloned Sr genes in wheat has been documented making isolation of *Sr26* and *Sr61* a valuable addition for inclusion into future transgenic cassettes.

However, transgene cassette deployment in wheat is currently constrained by GM regulations[39]. Here, we confirm the presence of both *Sr26* and *Sr61* in a recombinant alien introgression segment obtained by homologous recombination. This alien segment derived from tall wheat grass does not recombine with wheat chromosomes making it a combined *Sr26-Sr61* gene stack. In addition, this alien segment is considerably shorter than that in *Sr26*-bearing lines that have been used in commercial wheat cultivars, decreasing the likelihood of unwanted linkage drag. This alien gene stack containing both *Sr26* and *Sr61* offers a non-transgenic route for co-inheritance of both *R* genes and potentially increases the durability of both valuable resistance genes. However, that will occur only if the same genes are not exploited singly. *Sr26* is already deployed alone, but *Sr61* is currently present only in the presently reported lines.

## Methods

**Plant materials, mutagenesis, and mutant DNA preparation**. Wheat lines carrying *Sr26* (Avocet+Lr46) and *Sr61* (W3757) were mutagenised with EMS and progeny susceptible to *Pgt* race 34-1,2,3,4,5,6,7 were selected as described in a related study[22]. Genomic DNA was prepared from seedling leaves as described by Yu et al.[40]. DNA quality and quantity were assessed with a NanoDrop spectrophotometer (Thermo Fisher Scientific, Waltham, MA, USA) and by electrophoresis in 0.8% agarose gels.

**R gene enrichment and sequencing**. Target sequence enrichment of NLRs was undertaken by Arbor Biosciences (Ann Arbor, MI, USA) using the MYbaits protocol and the Triticeae NLR bait libraries Tv2 for *Sr26* and Tv3 for *Sr61*, available at https://github.com/steuernb/MutantHunter/blob/master/Triticea_RenSeq_Baits_V3.fasta.gz[8]. Library construction was undertaken using the TruSeq RNA Protocol v2. All enriched libraries were sequenced using a HiSeq 2500 (Illumina) sequencing platform that generated 250 bp paired-end reads. Sequencing data of wild-type and *Sr26* and *Sr61* mutants are summarized in Supplementary Data 2.

**Sequence analysis**. CLC Genomics Workbench v9.0 (*Sr26*) and v10.0 (*Sr61*) (Qiagen, Hilden, Germany) were used for read quality control, trimming, and de novo assembly of wild-type reads using the following parameters; minimum contig length: 250, auto-detect paired distances, perform scaffolding, mismatch cost: 2, insertion cost: 3, deletion cost: 3, length fraction from 0.5 to 0.9, and similarity fraction 0.9–0.98. For mapping sequence reads from wild-type and mutants against the de novo wild-type assembly the following parameters were used; no masking, linear gap cost, length fraction 0.5–0.9, and similarity fraction 0.95–0.98. *Sr26* contigs containing mutations in each line were identified using the MutantHunter[8] pipeline with default parameters whereas the MuTrigo Python package (https://github.com/TC-Hewitt/MuTrigo) was used for SNP calling with default parameters to identify candidate contigs containing mutations in the *Sr61* mutants. For *Sr61* analysis the mutant M1 sequence was used for de novo assembly due to insufficient data obtained from the wild-type (W3757) (Supplementary Data 2).

**Gene sequence confirmation**. Total RNA was extracted using a PureLink™ RNA Mini Kit (Invitrogen, Carlsbad, CA, USA) as per the manufacturer's instructions. cDNA synthesis was performed as described by Clontech (Mountain View, CA, USA). Flanking gene sequences were first amplified by 5′ and 3′ RACE (rapid amplification of cDNA ends) and then by using a Universal GenomeWalker kit (Takara Bio, Mountain View, CA, USA). Nonsynonymous substitutions identified in mutants by RenSeq were confirmed by PCR amplification of mutant DNAs and Sanger sequencing (Supplementary Table 3). Exon–intron structures were confirmed by cDNA amplification and sequencing.

**Candidate gene confirmation by wheat transformation**. Sr26 and Sr61 gene candidates were introduced into wheat cultivar Fielder by Agrobacterium-mediated transformation using binary vector pVecBARII and phosphinothricin as a selective agent[41]. $T_0$ plants were transplanted to a growth cabinet (23 °C, 16 h light/8 h darkness) and inoculated with Pgt race 98-1,2,3,5,6+Sr50 at 7–10 days post-transplantation. Rust reactions were assessed 10–15 days post Pgt inoculation[28].

**Rust phenotyping and histological assessment**. Stem rust phenotyping of seedlings and adult plants in the greenhouse and field were as previously described[42–44]. Histological assessments of Pgt infection site sizes were as previously described by Ayliffe et al.[45]. Images were photographed using a CC12 digital camera and AnalySIS LS Research v2.2 software for analysis (Olympus Soft Imaging System, Japan).

**Phylogenetic analysis**. R gene protein sequences from the NCBI database (protein accession numbers are listed in Supplementary Data 1) were aligned using MUS-CLE and phylogenetic trees were constructed using the UPGMA program in MegaX[46]. Evolutionary distances were determined using the Neighbour-Joining method[47] with Poisson correction and units to show the number of amino acid substitutions per site. All positions that contained either gaps or missing data were removed from the analysis. Phylogenetic trees were annotated using ITOL (https://itol.embl.de).

**Protein structure predictions**. Coiled-coil domains in predicted Sr26 and Sr61 proteins were indentified using the COILS prediction program[48] (https://embnet.vital-it.ch/software/COILS_form.html) and the T-Coffee program Expresso (http://tcoffee.crg.cat/apps/tcoffee/do:expresso) was used for protein sequence alignment.

**Molecular cytogenetic characterization of lines containing Sr26 and Sr61**. Root tips were pre-treated and slides were prepared according to Zhang et al.[49]. Non-denaturing fluorescence in situ hybridization (ND-FISH) with oligonucleotide probes OligopSc119.2-1 and Oligo-pTa535-1 was used to identify individual chromosomes[50]. OligopSc119.2-1 and Oligo-pTa535-1 were labeled with 6-carboxyfluorescein (6-FAM) and 6-carboxytetramethylrhodamine (Tamra), generating green and red signals, respectively. Chromosomes were counterstained with 4′,6-diamidino-2-phenylindole (DAPI) (Invitrogen) and pseudocolored blue. Photomicrographs were taken with a Retiga EXi CCD (charge-coupled device) camera (QImaging, Surrey, BC, Canada) mounted on a Zeiss Axio Imager epi-fluorescence microscope. After stripping off the oligo probes, the same slides were used to further characterize the Sr26 translocation line Avocet+Lr46, Sr61 substitution line W3757 and recombinant 378/15 by genomic in situ hybridization (GISH) as described by Zhang et al.[49]. Total genomic DNA from Pseudoroegneria stipifolia (accession PI 314058) was labeled with biotin-16-dUTP (Roche Diagnostics Australia, Castle Hill, NSW) using nick translation. Unlabeled total genomic DNA of wheat was used as blocker at a ratio of 1:80 (probe: blocker). Signals were detected with fluorescein avidin DN (Vector Laboratories, Burlingame, CA, USA). Chromosomes were counterstained with DAPI and pseudocolored red.

**Characterization of T-DNA copy number by digital PCR**. The phosphinothricin (PPT)-selectable marker gene located near the T-DNA left border was used to estimate the transgene copy number. The PPT primer pair and PPT-probe (labeled with 5′FAM (6-fluorescein) and doubled-quenched with ZEN™ and Iowa Black Hole Quencher 1) were used according to Petrie et al.[51]. Genomic DNA (approximately 100 ng) was digested with EcoRI (New England Biolabs, Ipswich, MA, USA) in a final volume of 20 μL for 4 h at 37 °C. Samples were placed onto Droplet Generator QX200™ (Bio-Rad) and heat sealed. Amplifications were carried in C1000 Thermal Cycler (Bio-Rad) and reactions were run with the following cycles: 95 °C for 10 min followed by 40 cycles at 94 °C for 30 s; 59 °C for 1 min, then 98 °C for 10 min and maintained at 12 °C finally. A ramping rate of 2.5 °C/s in all temperature change steps was employed. Following amplification reaction, the plates were placed onto a QX200 Droplet Reader (Bio-Rad). Data was analysed using Quanta soft™ software (Bio-Rad).

**qRT-PCR for determining transgene expression**. Leaf tissues from each sample were frozen in liquid nitrogen or dry ice immediately after sampling. RNA was isolated using a RNeasy® Plant Mini Kit (QIAGEN, Chadstone Center, VIC, Australia) according to the manufacturer's protocol. One to two microgram of RNA samples were used for first-strand DNA synthesis in 20 μL reactions using

Superscript® III reverse transcriptase kit (Life Technologies, Mulgrave, VIC, Australia). After the reverse transcript reaction, 3 μL of 10× dilutions of synthesis product were used for qPCR reaction using a C1000 Touch™ thermocycler with the CFX96™ Real-Time System (Bio-Rad). qPCR conditions included an initial denaturation at 95 °C for 3 min; 40 cycles of denaturation at 95 °C for 10 s and annealing/elongation at 60 °C for 30 s, followed by a melt step range of 65–95 °C with an increment of 0.5 °C. We used the wheat housekeeping gene TaCON as a reference gene for each qRT-PCR experiment[52]. qPCR primers specific for Sr61 (Sr61GSPF1 and Sr61GSPR1) were used to measure relative gene expression (Supplementary Table 3). Experiments included three technical replicates for each of three biological replications. δCq mean values were calculated and standard errors were determined. Gene expression values were log (base 2)-transformed.

**Reporting summary**. Further information on research design is available in the Nature Research Reporting Summary linked to this article.

## Data availability

Data supporting the findings of this work are available within the paper and its Supplementary Information files. A reporting summary for this article is available as a Supplementary Information file. The datasets and plant materials generated and analyzed during the current study are available from the corresponding authors upon request. Annotated genomic sequences of Sr26 and Sr61 have been deposited at NCBI GenBank with accession numbers MN531843 (Sr26) and MN531844 (Sr61). The source data underlying Supplementary Figs. 3f, 4, and 7c are provided as a Source Data file. Source data are provided with this paper.

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

## Acknowledgements

This work was supported by National Science Foundation, Basic Research Enabling Agriculture Development (NSF-BREAD), USA, and the Grains Research and Development Corporation (GRDC), Australia. The first author was supported by the Monsanto Beachell-Borlaug International Scholars Programs (MBBISP), USA, and the Research Training Program (RTP) of the Australian Department of Education and Training.

## Author contributions

J.Z. performed all the experiments to clone *Sr26* and *Sr61* and analysed the data. *Sr26* and *Sr61* mutants were generated by P.Z., R.A.M., S.H., W.S., and E.L. Rust pathology work in South Africa, Denmark, and Australia was conducted by W.H.B., Z.A.P., M.P., M.H., P.Z., R.A.M., J.Z., and R.F.P. NGS data analysis for candidate gene identification of *Sr26* was by T.C.H., N.U., SP., and J.Z., and for the *Sr61* candidate was carried out by J.Z. Design and testing of gene-specific markers was by J.Z., S.C., and E.L. J.Z., D.B., S.C., and M.L. performed the transgenic validation of *Sr26* and *Sr61* with oversight by M.A. Molecular cytogenetic experiments to characterize the wild-type and recombinant lines combining *Sr26* and *Sr61* were carried out by J.L. and P.Z. Phylogenetic tree was constructed and analysed by J.Z. and P.D. Initial development and characterization of recombinants were done by I.D. NLR bait libraries were developed and provided by B.B. H.W. and B.S. E.L., P.Z., and R.A.M. conceived, designed, and supervised the research. J. Z., E.L., R.A.M., P.Z., and M.A. drafted the manuscript and all co-authors provided edits.

## Competing interests

Patent applications based on this work have been filed by E.L., P.Z., and J.Z. and the filing reference numbers for *Sr26* and *Sr61* are PCT/AU2019/050331 and PCT/AU2020/051224, respectively. Other authors claim no competing interests.
