## [Peer Review File · Nature Communications]

Reviewers' comments:

Reviewer #1 (Remarks to the Author):

This manuscript describes the identification of the gene sequences for Sr26 and Sr61 using a mutational genomics approach. Stem rust is a serious disease in regions where virulent races are present and the results of this study will be of interest to a broad research community. The manuscript is very well written. In my opinion, the manuscript should be accepted with minor revisions.

I think the story is complete but some readers may ask why transgenic complementation was performed for Sr26 but not for Sr61. I don't see this as an issue but the editor may ask for an explanation. The colours used in Fig 3a make some of the text difficult to read, particularly the red text on the green background. Consider revising your colour choices.

A few specific comments/suggestions:

On the title page, the superscript affiliation numbers by the author names are not separated by commas for multiple affiliations.

Page 2 – “.....is one of the earliest examples of a successful transfer.....”

Page 3 – “.....carried a single or both gene(s).”

Page 5 – “.....were phenotyped with each Pgt race.”

Reviewer #2 (Remarks to the Author):

The manuscript described experiments on cloning two stem rust resistance genes, Sr26 and Sr61, that originated from tall wheat grass genome. These two genes are old, but still effective to current dominant races of stem rust, especially to Ug99, therefore they are important genes for stem rust control. Because homologous recombination does not occur between wheat chromosomes and tall wheat grass segments, map-based cloning approach was not used. Instead, a relatively new approach, MutRenSeq was used to identify candidates for the two resistance genes. Gene capture sequencing the mutants found two candidates for the two genes, respectively. Transgenic complementation experiments confirmed the candidate genes as causal genes for Sr26 and Sr61. Since these genes are alien segments, they developed dominant markers for each of the two genes for marker-assisted selection. The manuscript is well written, and experiments were straight forward. Followings are some additional minor points that may need to be addressed:

1. Abstract appears to have too much background information, too little result.
2. P3,L2: both genes
3. For mutant screening, it may be better to indicate the number of plants screened to get these mutants for both genes.
4. P3, L13: Here reported 5 mutants, but 6 mutants were reported in a previous paper from the same experiment. Is any reason for that?
5. P3, L13: a linked marker. Is this marker developed from a previous study? If so, please specify. Also, based on the published information and information from this study, this is relatively long alien segment, a linked marker previously developed could be any DNA sequence in the segment. Does missing the marker means missing entire alien segment? Is any information on how far this marker from the gene? The same question needs to be addressed for Sr61.
6. P3,L21: Here 'which' means all 3 or 2 of the 3 mutations from the same event? Please clarify.
7. P4,L24: Minimal flanking region?

8. Fig S3: Missing Fielder control as indicated in Figure legend.
9. P5-L3, both....lines.
10. I understand that the two markers were sequences from alien segments, but sometimes they are not genome specific, thus suggest to genotype a set of breeding materials/or cultivars to validate if the markers are really diagnostic.
11. For Sr26 markers, 1.5kb is too long. Since it is from alien segment and a dominant marker, it is better to be shortened to 2-300bp.
12. For transgenic work, suggest add a supplemental material to describe construct sequences used for transformation.
13. Fig. S1: "Vertical bars represent the positions of the SNPs", but no such bars can be found in the Figure, instead there are lines of two colors. Please fix them.

Reviewer #3 (Remarks to the Author):

Comments from reviewer:

General comments:

This manuscript reported a significant contribution to wheat improvement for stem rust resistance by isolating Sr26 and Sr61, two of the very few R genes to which virulence has not been observed in Pgt populations, and by identifying and characterizing a recombinant of Sr26 and Sr61. The study is an extension of the early study by Mago et al. (ref# 20) that created the critical germplasm of a recombinant of these two important stem rust resistance genes. The alien nature of this recombinant segment, identified and characterized in the manuscript through gene isolation, is expected to remain as a single inheritance unit during the subsequent breeding process to ensure their co-deployment. The ease of inducing recombination of the Thinopyrum chromosome segments in wheat background, demonstrated in the early study (ref #20) and characterized in this study, opened a potential avenue for incorporating additional useful genes from Thinopyrum, including other R genes, into this recombinant segment. As stated by the authors, this approach bypasses the stigma of "GMO" for combining and co-deploying multiple R genes, as encountered by "gene-cassette" approach for the same purpose. Although isolating the genes for the purpose of identification appears to be a long route approach, the objective was achieved as was intended. The study further demonstrated that MutRenSeq is a powerful tool in cloning R genes, enabling gene isolation from germplasm that nearly out-of-reach by conventional map-based cloning approach.

Specific comments/suggested changes:

1. High infection types on upper leaf surface?

"Figure 1 Rust phenotypes of ... (a) Lower and upper seedling leaf surface of wild type ..." Contrary to what the title said, it is doubtful to this reviewer that any upper leaf surface was shown on the infection type panel of Figure 1, at least for those with high infection types to PTKST, such as Mutant 12S, Mutant M4, and 37-07. Two leaf surfaces of primary leaves of wheat, upper (or adaxial) and lower (abaxial) surfaces, normally react differently to stem rust infection. For a given uredinium, it penetrates and appears on both sides of a leaf. However, the size of uredinia on the lower leaf surface is always larger and more consistent in shape. Thus, the description of a typical infection type for a given Sr gene stock, or scoring stem rust infection types at the seedling stage, is generally based on the uredinia morphology on lower (abaxial) leaf surface. By the way, adaxial vs abaxial might be more appropriate to use in such journal as Nat. Comm.

2. Should isolate name be required?

A race name will not remain static and will change sooner or later once a nomenclature system is updated or changed. For instance, an isolate (04KEN156/04), identical in virulence profile to the original Ug99, was keyed to a race called TTKS. When nomenclature changed (by addition of a fifth set of differential lines Sr24, 31, 38 and McN), TTKS split into multiple races, and the isolate

04KEN156/04 was keyed to race TTKSK. Furthermore, we know that two of the races used in this study, TKTTF and TTKTT, have different variants in each. One type of TKTTF is avirulent on Sr7a and Sr45 and the other type is not. One variant of TTKTT is virulent on Sr8155B1 and the other type is avirulent. Thus, a permanent identity for the rust cultures used in the study should be given, and the use of a race name for this purpose is not going to work. This can easily be accomplished by requiring addition of an isolate name for each rust culture used. Such addition will be essential to ensure the identification of correct rust cultures in the future for conformational or follow-up studies, and will encourage the preservation of reference cultures published in refereed journals, especially prestigious journals like Nat. Comm. I realize that Nat. Comm. has published stem rust papers with race name only, but for the betterment of our science, we should not keep propagating this mistake any further.

3. Virulence of TTRTF on Sr33?

"The causal race (TTRTF) of the Sicilian...is virulent on seedlings carrying Sr35 and putatively Sr33...". Our lab has done confirmation experiments that CSID 5405 (stock for Sr33) tested with three different isolates of race TTRTF (from Georgia, Italy, and Hungary) showed infection types 2 to 2+, and transgenic of Sr33 (PC1(Sr33)ST3R1) showed infection type 2-. The results have been communicated to one of the corresponding authors as well as GRRC. An earlier stock representing Sr33 distributed from CDL to some of the workers in the author list was incorrect. I want to make sure that the statement "putatively virulent on Sr33" was not based on the incorrect stock. I encourage reconfirmation experiment to be done if the authors think "putatively virulent on Sr33" (of TTRTF) is an important point to be made here (which I do not think so), assuming that the correct stock of Sr33 was used. A more complete description of the virulence of race TTRTF (as well as the origin of this race) can be found in: Olivera et al. 2019 Phytopathology (<https://doi.org/10.1094/PHYTO-06-19-0186-R>)

4. Validation of the Sr61 candidate gene?

Was validation through transformation done for Sr61? If not, why not? Should the lack of validation be elaborated in the manuscript? I also wonder why mutants of Sr61 were not shown in Figure S7a,b,d.

5. Entries in table vs figure:

Table S3 listed as N/A for testing of Sr61 mutants to race TTRTF, and yet the phenotype photo Figure S6c showed at least three of the Sr61 mutants (M2, M4 and M6) were tested. Furthermore, tests with other two races, TTKTF and TKTTF, showed on Figure S6c, were not listed in Table S3. Should a table be more inclusive whereas a figure should show a subset of the entries for demonstration purpose?

6. Other resistance genes in W3757?

Three mutants of Sr61 remained resistant to TTRTF and TTKTF (Figure S6c). Does that indicate additional resistance gene(s) may be present in W3757 that remained effective to these races after Sr61 was knocked out through mutation? If so, could "virulence unknown for Sr61" be a result of multiple genes?

7. "S" response in a mutant line:

On the same image panel (Figure S6c), M2 displayed an infection type 2+ to TTKTT, whereas Table S3 listed M2 as "S" to TTKTT. Infection type of M2 (2+) is higher than W3757 (2-2), but they are not qualitatively different, and I would not consider that to be susceptible, especially in comparison with the susceptible check Morocco on the same image panel.

8. Other minor comments the authors may consider during their revision

- Correct race name from ref.# 11--P2-L13, should be races "TRTTF and JRCQC" instead of "TTRTF and JRCQC".
- Infection type on adult plant--As a common practice, infection type refers to reaction at the seedling stage (by notation of ; 1, 2 etc) and infection response refers to reaction at the adult plant stage (by

notation of R, MR etc.). "Infection types on flag leaves", although mentioned in the reference cited (#33), was never adequately defined or diagrammatically described, and has not been routinely used in the measurement of stem rust resistance. Two independent occasions in this manuscript, "High infection type on adult plant" (P2L14-15) and Figure S6, have used infection types at the adult plant stage. Was the intention of the authors to use Figure S6b to define "flag leaf infection type"? Why infection responses instead of flag leaf infection types were used in Figure S7d if definitions have been given in Figure S6b?

- Rust phenotyping protocols across different labs--Based on the rust cultures used and authors, it is apparent that the pathology work was done in three different labs (PBI, UFS, GRRRC). I am aware that some variation exists between different labs, such as dpi, primary or second leaf to be used in the assessment of infection type, etc. References citation for rust phenotyping (#32, 33) may not cover all the protocols deployed in this research. In particular, dpi was not given in Figure S6 © as were done in (a) and (b).
- Infection type definition on Figure S6c—I wonder if the authors could revisit the infection type ratings on some of the lines on Figure S6c. It is a bit tenuous to classify line M4 (tested with TTRTF) or W3757 (tested with TTKTT) into the infection type 2 category when green island is obviously absent, whereas Avocet to TTKTT was classified as IT ;1- when green island was obvious, thus should be in the IT 2 category.
- Figure S6 did not include W3757 (Sr61+) in the adult plant test with PTKST. Figure 1 and Figure S7 should be cited to support the statement of P5L6-8 instead of Figures S6 and S7 cited. I wonder what the infection response to PTKST would be as seedling of W3757 had an "X-" (Figure 1a) instead of a ";" as expected for Sr61 (shown in ref #20). In fact, W3757 was substantially higher than ";" to all other races tested (see Figure S6c).
- The color of uredinia of Mutant M4 (Figure 1) is a bit unusual/unreal for stem rust, a stark contrast with other images of the same race. Try to substitute with a different photo of more realistic color if available.
- Pathotype instead of race was used in (supplementary Figure S6) whereas race was used in most of the manuscript.
- The susceptible control showed in Figure S3 was labeled as "Non-transformed sib line", whereas the title of the figure listed as control Fielder. I do not agree the term 'sib' to be used here.
- Rust reaction of individual T1 plants from selected T0 families were displayed for two constructs: Fielder: Sr26: Sr22RE and Fielder: Sr26: Sr33RE in Figure S3a and S3b, but not the third construct Field: 26: NativeRE. Were T1 plants of Field: 26: NativeRE assayed for rust reaction?
- Reference #20 was published in TAG in 2019, not 2018.
- Was Pgt race (presumably 34-1,2,3,4,5,6,7) used to generate data for Table S1 and Table S2? The race name should be given regardless.
- The title of Table S3 should be "Multiple Pgt races used in the tests of Sr26 and Sr61 wildtype and mutants", instead of "multiple Pgt tests...".
- Table S3 would be a good place to supply the isolate identity (as I have commented above#2), as was done for Pgt race TTKTT in the table. I also recommend the inclusion of infection response (or infection type) data for race TTKTF, and TKTTF that were shown on Figure S6c (and commented above).

- I have difficulties to understand the choice of the word "Resolving" in the title, ambiguous to me.

Reviewer: Yue Jin

Review completed Nov 15, 2019

Reviewer #4 (Remarks to the Author):

Jianping Zhang et al. studied wheat stem rust resistance genes, Sr26 and Sr61. Both genes are originated from the wheat relative - *Thinopyrum ponticum*, which demanded a unique cloning approach when the work is done in wheat. Authors used the RenSeq approach and identified two NLR genes that were effective for stem rust resistance in wheat. Cytogenetic studies illustrated a small introgressed Ae segments with Sr26 and Sr61 in wheat. The stack of both genes in one chromosome segment enables the co-deployment of both genes in cultivated wheat. This study generated useful data for wheat community, the manuscript is well written, and conclusions are partially convincing. However, some results and conclusions require further clarification and more experimental data.

Major comments

1. "... from tall wheat grass" in the title is misleading. Readers may think both genes were directly cloned from the tall wheat grass. Rephrase the title to make it nonambiguous.
2. Authors indicated the broad-spectrum wheat stem rust resistance of Sr26 and Sr61. How many Pgt races were tested for each gene? Will the tested Pgt races represent a broad-spectrum of virulence of all current Pgt races worldwide? Authors need clarify how these genes confer a broad-spectrum resistance to Pgt or delete the concept.
3. Authors stated that "Whether or not the durability of Sr26 and Sr61 were due to a combined effect from multiple R genes located on the introgressed *Th. ponticum* alien segment remain unknown." If this is the case, how can they be certain that the cloned R genes are Sr26 and Sr61?
4. MutRenSeq was used to clone both genes. Apparently, authors assumed that Sr26 and Sr61 are both NLR genes. As discussed in the paper, the region may contain a cluster of resistance genes. Is it possible that the cloned genes are not the true Sr26 and Sr61 genes? They will need clarify in the manuscript what drove them to believe that Sr26 and Sr61 must be NLR genes and then pursued MutRenSeq.
5. Authors stated that "We identified five susceptible ethyl methanesulfonate (EMS)-induced mutants from the Sr26-carrying wheat genetic stock, Avocet+Lr46 We also identified eight susceptible EMS mutants derived from line W3757 thereby enabling isolation of Sr61." Authors should tell the mutant population sizes of Avocet+Lr46 and W3757, numbers of susceptible mutants, and numbers of mutant lines with non-synonymous substitutions or deletions in the two cloned NLR genes.
6. Apparently, authors believe that the two NLRs represent Sr26 and Sr61 and they stated like this in the manuscript. Is it possible that there are other resistance genes in the designated locus?
7. "In the current study, none of the three mutations in the LRR domain of Sr61 resulted in a new resistance specificity based on the Pgt races tested." What has been done to make this statement?
8. What is the purpose of structure modelling in Figure S2b? Does this help resolve the mechanisms of the mutation? If not, they should delete modelling.
9. Authors proved that T1 plants of Fielder: Sr26: Sr22RE and Fielder: Sr26: Sr33RE were resistant to the Pgt race 98-1,2,(3),(5),6. What is the result of T1 plants of Fielder: Sr26: NativeRE? This is important data but missing in the manuscript. Readers want to know whether the native Sr26 alone (viz. 917bp + 6066bp + 263bp) is sufficient to confer an inheritable resistance.
10. Agrobacterium mediated transfer can generate single or low-copy insertions. What is the copy number of the transgene in selected T1 families? In their T1 test with Pgt race 98-1,2,(3),(5),6, what is the segregation ratio between resistance and susceptibility? Is there a correlation between copy numbers, cDNA levels and stem rust severity? The data should be provided to help readers understand how the gene works.
11. Author stated that "Thus the gene candidate was not only necessary but also sufficient to confer Sr26 resistance (Tables S1, S3)." Have they compared the resistance spectrum of transgenic plants

and the Sr26 donor? If not, how do they know the candidate gene confer the Sr26 resistance?

12. Given that the additional Sr22 and Sr33 promoter and terminator sequences did not abolish Sr26 gene function, does the presence of Sr22 and Sr33 RE sequences boost the Sr26 gene function?

13. Unfortunately, we don't see the complementation data of the Sr61 candidate. Although the complementation does not necessarily tell the identified NLR is Sr61, it will support that the identified NLR is effective for stem rust resistance. A complementation experiment should be done and included for the Sr61 candidate.

14. In Figure S4, the PCR products did not migrate at the same line. Could this be caused by amplifying different alleles? Did they sequence the PCR product to confirm their identity? Which Sr26+Sr61 recombinant (+) was used in S4a, 378/15, 388/15, or 376/15?

15. In Figure S6c, it appears that both W3757 behaved similar to Sr61 M2 in response to TTKTF, TTKTT, and TTRTF. IF this is true, the mutation of the Sr61 candidate did not compromise the resistance to these races. Another gene should be involved in conferring resistance in the W3757 background.

16. Wheat has three type of rust problems. In the phylogenetic tree, both stem rust and leaf rust R genes are analyzed. As reported, Yr5 and YrAS2388 are also NLR genes, are these genes included in this tree? Any other yellow rust resistance genes should be included here? It is interesting to see how all wheat rust resistance genes are classified in one tree.

17. To provide the source of all accessions in Figure 3a, it is useful to add species (eg. Os, Hv, and Ae) as prefix for each accession and explain the prefix in legend. In Data S1, add two columns, for species and diseases, respectively.

18. This study misses the expression profiles of both genes, regarding to the developmental stages, tissue specificity, and pathogens. The expression profiles are necessary for understanding the regulation of these genes.

19. Because the two NLR genes were effective, they might discuss how to manage gene stewardship to increase resistance durability. They might also discuss the fate of the two NLR genes after they are deployed in large area in regions with severe stem rust epidemics.

Minor comments

1. Some affiliations miss postcodes.

2. In the introduction, they stated that "Sr26 was transferred to wheat chromosome 6A ... and this resistance has remained effective against all known Pgt races.... to date no known virulence has been reported for Sr61." This again is related to the wide-spectrum term. It will be helpful to introduce how the two genes were deployed in production, such as how many years on how much hectares.

3. Add a period sign after "... deletion of a linked marker (Figure1a; Table S1)"

4. Italicize Lr46 in "Avocet+Lr46".

5. Add space in "Figure1b" in "LRR motifs at the C-terminus (CNL) (Figure1b)"

6. Delete "translated" from "and effects in the predicted translated Sr26 and Sr61 proteins" in the Figure 1 legend.

7. In Figure 1, use a simple straight line to replace the while doted blocks (introns). The NB-ARC motif should exclude the intron region.

8. In Figure 1b, 499S was incorrectly placed.

9. Should it be "S1b" in "were analysed by NLR gene capture and sequencing (Figures 1a, S1a)."

10. "Sr61 M1" is missing in Figure S2a.

11. Authors stated that "A single contig of 3,519 bp was identified that contained nucleotide changes in five of the mutants (Figure S1b) identified an additional alteration in the sixth mutant (M6)." Supply IDs for the first five mutants and delete "additional".

12. In Figure S2a, a note is required for the blue frame in the Sr26 sequence prior to the residue 486 requires.

13. In Figure 2a. "Sr26 ORF" should be changed to "Sr26 ORF-corresponding genomic region"

14. In Figure 2b, add labels to the picture directly, instead of using 1 to 7.

15. In Figure S3a, missing susceptible control Fielder. "(A) .. (B)..." should be changed to small case.

16. In Table S1, progeny test (M3) must be done for the S sib, but missing this specific information. This manuscript used M1 to M8 as IDs for Sr61, which are confusing with the mutant generation

numbers. To specify a generation, it is better use M with numbers as subscripts. A note of SrB = Sr61 should be added, or replacing SrB with Sr61. Clarify if the R sib (M2) and Mutant (M2) were derived from the same M1 plant.

17. In Figure S7a-b, d, explain how to understand the disease severity scores, such as 20MRR; in Figure S7c, revise "Sr26mutant 12S1"

18. Use "3b" in "Sr13 (58.46% identity at protein level) (Figure 3B)"; delete ", 3b" in "barley Mla R gene family (Figures 3a, Clade I, 3b). "

19. In Table S2, add notes or reference to clarify infection type scores; explain how different scores for the same event were collected, eg. 1+2-2; provide infection types for T1 generations.

20. In Table S3, specify "Avocet" used.

Reviewers' comments:

Reviewer #1 (Remarks to the Author):

This manuscript describes the identification of the gene sequences for Sr26 and Sr61 using a mutational genomics approach. Stem rust is a serious disease in regions where virulent races are present and the results of this study will be of interest to a broad research community. The manuscript is very well written. In my opinion, the manuscript should be accepted with minor revisions.

I think the story is complete but some readers may ask why transgenic complementation was performed for Sr26 but not for Sr61. I don't see this as an issue but the editor may ask for an explanation. The colours used in Fig 3a make some of the text difficult to read, particularly the red text on the green background. Consider revising your colour choices.

Response:

We have now produced 14 independent transgenic wheat lines that contain a *Sr61* wheat transgene. These transgenics show a similar resistance phenotype to lines containing the endogenous gene and show similar levels of expression. This is a substantial addition to the manuscript and removes all possible doubt about the identity of the *Sr61* gene.

We have revised the colours in Figure 3 as suggested.

A few specific comments/suggestions:

On the title page, the superscript affiliation numbers by the author names are not separated by commas for multiple affiliations.

Response:

Revised accordingly.

Page 2 – “....is one of the earliest examples of a successful transfer.....”

Response:

Revised accordingly.

Page 3 – “....carried a single or both gene(s).”

Response:

Revised accordingly.

Page 5 – “.....were phenotyped with each Pgt race.”

Response:

Revised accordingly.

Reviewer #2 (Remarks to the Author):

The manuscript described experiments on cloning two stem rust resistance genes, Sr26 and Sr61, that originated from tall wheat grass genome. These two genes are old, but still effective to current dominant races of stem rust, especially to Ug99, therefore they are important genes for stem rust control. Because homologous recombination does not occur between wheat chromosomes and tall wheat grass segments, map-based cloning approach was not used. Instead, a relatively new approach, MutRenSeq was used to identify candidates for the two resistance genes. Gene capture sequencing the mutants found two candidates for the two genes, respectively. Transgenic complementation experiments confirmed the candidate genes as causal genes for Sr26 and Sr61. Since these genes are alien segments, they developed dominant markers for each of the two genes for marker-assisted selection. The manuscript is well written, and experiments were straight forward.

Followings are some additional minor points that may need to be addressed:
1. Abstract appears to have too much background information, too little result.

Response:

We have rewritten the abstract and included more detail on the gene products that were identified and the importance of these findings.

2. P3,L2: both genes

Response:

Revised accordingly.

3. For mutant screening, it may be better to indicate the number of plants screened to get these mutants for both genes.

Response:

Revised accordingly. The numbers of plants used for both mutation series were added in the text.

4. P3, L13: Here reported 5 mutants, but 6 mutants were reported in a previous paper from the same experiment. Is any reason for that? (can be explained as either lacking seeds of the Six mutants or we thought 4 mutants with 1 deletion would be sufficient for MutRenSeq)

Response:

We felt that 4 point mutations and 1 deletion would be sufficient for gene identification, given MutRenSeq is an expensive undertaking for each sample. As we have generated Sr26 transgenic wheat lines and confirmed the gene function there can be no doubt about its identity and hence the number of mutants initially analysed by MutRenSeq proved to be sufficient for gene identification.

5. P3, L13: a linked marker. Is this marker developed from a previous study? If so, please specify. Also, based on the published information and information from this study, this is relatively long alien segment, a linked marker previously developed could be any DNA sequence in the segment. Does missing the marker means missing entire alien segment? Is any information on how far this marker from the gene? The same question needs to be addressed for Sr61.

Response:

We have added the marker names and references for both genes. The reviewer is correct, we also predicted that loss of the linked marker could be indicative of loss of a relatively long segment or possibly the entire chromosome arm. Unfortunately, there is no accurate information related to the physical distance from the linked marker to either *Sr26* or *Sr61* as there isn't sequence information available for either introgressed *Th. ponticum* segment.

6. P3,L21: Here 'which' means all 3 or 2 of the 3 mutations from the same event? Please clarify.

Response:

Revised accordingly.

7. P4,L24: Minimal flanking region?

Response:

We have rewritten the sentence.

8. Fig S3: Missing Fielder control as indicated in Figure legend.

Response:

Revised accordingly.

9. P5-L3, both....lines.

Response:

Revised accordingly.

10. I understand that the two markers were sequences from alien segments, but sometimes they are not genome specific, thus suggest to genotype a set of breeding materials/or cultivars to validate if the markers are really diagnostic.

Response:

These results were added in revised Figure S4.

11. For *Sr26* markers, 1.5kb is too long. Since it is from alien segment and a dominant marker, it is better to be shortened to 2-300bp.

Response:

We made a huge effort to develop a shorter marker for *Sr26* for the reason pointed out by the reviewer; however, we have tested around 40 pairs of shortened markers and all failed to show sufficient specificity. This difficulty is invariably a result of the multiple copies of related *Sr26* sequences in group 6 chromosomes of the wheat genome. In the interim the 1.5kb amplicon remains an accurate predictor of the presence of *Sr26*. We plan to continue to work on this in order to provide a better breeder-friendly marker.

12. For transgenic work, suggest add a supplemental material to describe construct sequences used for transformation.

Response:

All construct sequences are being deposited into NCBI together with both gene sequences.

13. Fig. S1: "Vertical bars represent the positions of the SNPs", but no such bars can be found in the Figure, instead there are lines of two colors. Please fix them.

Response:

Revised accordingly.

Reviewer #3 (Remarks to the Author):

Comments from reviewer:

General comments:

This manuscript reported a significant contribution to wheat improvement for stem rust resistance by isolating Sr26 and Sr61, two of the very few R genes to which virulence has not been observed in Pgt populations, and by identifying and characterizing a recombinant of Sr26 and Sr61. The study is an extension of the early study by Mago et al. (ref# 20) that created the critical germplasm of a recombinant of these two important stem rust resistance genes. The alien nature of this recombinant segment, identified and characterized in the manuscript through gene isolation, is expected to remain as a single inheritance unit during the subsequent breeding process to ensure their co-deployment. The ease of inducing recombination of the Thinopyrum chromosome segments in wheat background, demonstrated in the early study (ref #20) and characterized in this study, opened a potential avenue for incorporating additional useful genes from Thinopyrum, including other R genes, into this recombinant segment. As stated by the authors, this approach bypasses the stigma of “GMO” for combining and co-deploying multiple R genes, as encountered by “gene-cassette” approach for the same purpose. Although isolating the genes for the purpose of identification appears to be a long route approach, the objective was achieved as was intended. The study further demonstrated that MutRenSeq is a powerful tool in cloning R genes, enabling gene isolation from germplasm that nearly out-of-reach by conventional map-based cloning approach.

Specific comments/suggested changes:

1. High infection types on upper leaf surface? “Figure 1 Rust phenotypes of ... (a) Lower and upper seedling leaf surface of wild type ...” Contrary to what the title said, it is doubtful to this reviewer that any upper leaf surface was shown on the infection type panel of Figure 1, at least for those with high infection types to PTKST, such as Mutant 12S, Mutant M4, and 37-07. Two leaf surfaces of primary leaves of wheat, upper (or adaxial) and lower (abaxial) surfaces, normally react differently to stem rust infection. For a given uredinium, it penetrates and appears on both sides of a leaf. However, the size of uredinia on the lower leaf surface is always larger and more consistent in shape. Thus, the description of a typical infection type for a given Sr gene stock, or scoring stem rust infection types at the seedling stage, is generally based on the uredinia morphology on lower (abaxial) leaf surface. By the way, adaxial vs abaxial might be more appropriate to use in such journal as Nat. Comm.

Response:

Revised accordingly. The “lower and upper” we used here was to describe the primary and secondary leaves respectively. To avoid misunderstanding, we replaced the “Lower and upper” with “Abaxial” in Figure 1.

2. Should isolate name be required?

A race name will not remain static and will change sooner or later once a nomenclature system is updated or changed. For instance, an isolate (04KEN156/04), identical in virulence profile to the original Ug99, was keyed to a race called TTKS. When nomenclature changed (by addition of a fifth set of differential lines Sr24, 31, 38 and McN), TTKS split into multiple races, and the isolate 04KEN156/04 was keyed to race TTKSK. Furthermore, we know that two of the races used in this study, TKTTF and TTKTT, have different variants in each. One type of TKTTF is avirulent on Sr7a and Sr45 and the other type is not. One variant of TTKTT is virulent on Sr8155B1 and the other type is avirulent. Thus, a permanent identity for the rust cultures used in the study should be given, and the use of a race name for this purpose is not going to work. This can easily be accomplished by requiring addition of an isolate name for each rust culture used. Such addition will be essential to ensure the identification of correct rust cultures in the future for conformational or follow-up studies, and will encourage the preservation of reference cultures published in refereed journals, especially prestigious journals like Nat. Comm. I realize that Nat. Comm. has published stem rust papers with race name only, but for the betterment of our science, we should not keep propagating this mistake any further.

Response:

We have added a separate supplementary Table S6 to list all names of the isolates used in this study as requested.

3. Virulence of TTRTF on Sr33?

“The causal race (TTRTF) of the Sicilian...is virulent on seedlings carrying Sr35 and putatively Sr33...”. Our lab has done confirmation experiments that CSID 5405 (stock for Sr33) tested with three different isolates of race TTRTF (from Georgia, Italy, and Hungary) showed infection types 2 to 2+, and transgenic of Sr33 (PC1(Sr33)ST3R1) showed infection type 2-. The results have been communicated to one of the corresponding authors as well as GRRC. An earlier stock representing Sr33 distributed from CDL to some of the workers in the author list was incorrect. I want to make sure that the statement “putatively virulent on Sr33” was not based on the incorrect stock. I encourage reconfirmation experiment to be done if the authors think “putatively virulent on Sr33” (of TTRTF) is an important point to be made here (which I do not think so), assuming that the correct stock of Sr33 was used. A more complete description of the virulence of race TTRTF (as well as the origin of this race) can be found in: Olivera et al. 2019 Phytopathology (<https://doi.org/10.1094/PHYTO-06-19-0186-R>)

Response:

We have followed the suggestion of the reviewer and reference of virulence to *Sr33* has been removed.

4. Validation of the Sr61 candidate gene?

Was validation through transformation done for Sr61? If not, why not? Should the lack of validation be elaborated in the manuscript? I also wonder why mutants of Sr61 were not shown in Figure S7a,b,d.

Response:

We have now produced 14 independent transgenic wheat lines that contain a *Sr61* wheat transgene. These transgenics show a similar resistance phenotype to lines containing the endogenous gene and show similar levels of expression. This is a substantial addition to the manuscript and removes all possible doubt about the identity of the *Sr61* gene.

Sr61 mutants were not shown in Figures S7a, b, d as this material was unavailable at the time as *Sr26* mutants were identified. None-the-less we have included this figure as the *Sr26* data

is obviously valuable. In addition, we do however clearly show *Sr61* mutant phenotypes in Figure S6a and c.

5. Entries in table vs figure:

Table S3 listed as N/A for testing of Sr61 mutants to race TTRTF, and yet the phenotype photo Figure S6c showed at least three of the Sr61 mutants (M2, M4 and M6) were tested. Furthermore, tests with other two races, TTKTF and TKTF, showed on Figure S6c, were not listed in Table S3. Should a table be more inclusive whereas a figure should show a subset of the entries for demonstration purpose?

Response:

The reviewer is absolutely correct, and we have amended this oversight by adding this data to what is now Table S5 (formerly S3).

6. Other resistance genes in W3757? M₂

Three mutants of Sr61 remained resistant to TTRTF and TTKTF (Figure S6c). Does that indicate additional resistance gene(s) may be present in W3757 that remained effective to these races after Sr61 was knocked out through mutation? If so, could “virulence unknown for Sr61” be a result of multiple genes?

Response:

The reviewer is correct on this point and we have addressed this concern extensively above in the opening comments to the editor. Rather than re-iterate these points we direct the reviewer to comments above to the editor to address this point.

7. “S” response in a mutant line:

On the same image panel (Figure S6c), M2 displayed an infection type 2+ to TTKTT, whereas Table S3 listed M2 as “S” to TTKTT. Infection type of M2 (2+) is higher than W3757 (2-2), but they are not qualitatively different, and I would not consider that to be susceptible, especially in comparison with the susceptible check Morocco on the same image panel.

Response:

We fully agree with the reviewer that the background resistance in W3757 does make scoring a little difficult. Certainly, it would have been ideal to have a gene like Sr61 introduced into a background similar to Morocco to avoid confounding effects of other genes. However, we believe there is more growth on the 3 mutant lines for isolate TTKTT than the wild type W3757 line such that it is likely that this isolate is also recognised by the Sr61 gene we have cloned. Regardless the gene we have cloned still accounts for a significant (at least 4 other races) and, possibly all, the resistance encoded on the 6AE segment in W3757 and is justifiably consider as Sr61. Further analysis of this gene in other genetic backgrounds will ultimately enable the full resistance spectrum of this gene to be established. We revised the classification in Table S3 (now Table S5) accordingly.

8. Other minor comments the authors may consider during their revision

- Correct race name from ref.# 11--P2-L13, should be races “TRTTF and JRCQC” instead of “TTRTF and JRCQC”.

Response:
Revised accordingly.

- Infection type on adult plant--As a common practice, infection type refers to reaction at the seedling stage (by notation of ; 1, 2 etc) and infection response refers to reaction at the adult plant stage (by notation of R, MR etc.). “Infection types on flag leaves”, although mentioned in the reference cited (#33), was never adequately defined or diagrammatically described, and has not been routinely used in the measurement of stem rust resistance. Two independent occasions in this manuscript, “High infection type on adult plant” (P2L14-15) and Figure S6, have used infection types at the adult plant stage. Was the intention of the authors to use Figure S6b to define “flag leaf infection type”? Why infection responses instead of flag leaf infection types were used in Figure S7d if definitions have been given in Figure S6b?

Response:

We have revised the “High infection type on adult plant” to “infection response” and changed the scores of Figure S6b.

- Rust phenotyping protocols across different labs--Based on the rust cultures used and authors, it is apparent that the pathology work was done in three different labs (PBI, UFS, GRRC). I am aware that some variation exists between different labs, such as dpi, primary or second leaf to be used in the assessment of infection type, etc. References citation for rust phenotyping (#32, 33) may not cover all the protocols deployed in this research. In particular, dpi was not given in Figure S6c as were done in (a) and (b).

Response:

We added the dpi in Figure S6c and cited more protocols in the rust phenotyping and histological assessment section of the M&M as requested.

- Infection type definition on Figure S6c—I wonder if the authors could revisit the infection type ratings on some of the lines on Figure S6c. It is a bit tenuous to classify line M4 (tested with TTRTF) or W3757 (tested with TTKTT) into the infection type 2 category when green island is obviously absent, whereas Avocet to TTKTT was classified as IT ;1- when green island was obvious, thus should be in the IT 2 category.

Response:

We removed the infection type to avoid the potential confusion.

- Figure S6 did not include W3757 (Sr61+) in the adult plant test with PTKST.

Response:

Unfortunately, we do not have a photo for W3757 flag leaf inoculated by PTKST at the adult plant stage. But we do show the seedling stage inoculation of W3757 by PTKST in Figure 1a and added the adult plant stem of W3757 in Figure S7b.

- Figure 1 and Figure S7 should be cited to support the statement of P5L6-8 instead of Figures S6 and S7 cited.

Response:

Revised accordingly.

- I wonder what the infection response to PTKST would be as seedling of W3757 had an “X-” (Figure 1a) instead of a “;” as expected for Sr61 (shown in ref #20). In fact, W3757 was substantially higher than “;” to all other races tested (see Figure S6c).

Response:

The reviewer is correct and we also noticed the infection response of W3757 at the seedling stage was variable against different pathotypes. We agree that the infection type of W3757 is an “X-” against PTKST.

- The color of uredinia of Mutant M4 (Figure 1) is a bit unusual/unreal for stem rust, a stark contrast with other images of the same race. Try to substitute with a different photo of more realistic color if available.

Response:

Revised accordingly. The reason for the colour variation was as mentioned previously; the photo of W3757 and Mutant M4 were captured at a later batch of inoculation, therefore, it is difficult to get an exact tone match for the pictures. None-the-less we have changed the image contrast to the best of our ability as requested.

- Pathotype instead of race was used in (supplementary Figure S6) whereas race was used in most of the manuscript.

Response:

Revised accordingly.

- The susceptible control showed in Figure S3 was labeled as “Non-transformed sib line”, whereas the title of the figure listed as control Fielder. I do not agree the term ‘sib’ to be used here.

Response:

Revised accordingly.

- Rust reaction of individual T1 plants from selected T0 families were displayed for two constructs: Fielder:Sr26:Sr22RE and Fielder:Sr26:Sr33RE in Figure S3a and S3b, but not the third construct Field:26:NativeRE. Were T1 plants of Field:26:NativeRE assayed for rust reaction?

Response:

We have added images of the rust analysis done on T₁ transgenics containing the Sr26:NativeRE to the manuscript in Figure S3a, S3b and S3c as requested.

- Reference #20 was published in TAG in 2019, not 2018.

Response:

Revised accordingly.

- Was Pgt race (presumably 34-1,2,3,4,5,6,7) used to generate data for Table S1 and Table S2? The race name should be given regardless.

Response:

Pgt races were added to Table S1 and Table S2 as requested.

- The title of Table S3 should be “Multiple Pgt races used in the tests of Sr26 and Sr61 wildtype and mutants”, instead of “multiple Pgt tests...”.

Response:

Revised accordingly.

- Table S3 would be a good place to supply the isolate identity (as I have commented above#2), as was done for Pgt race TTKTT in the table. I also recommend the inclusion of infection response (or infection type) data for race TTKTF, and TKTTF that were shown on Figure S6c (and commented above).

Response:

A new table S5 has been included that specifies race name, isolate identity and where it is located as requested. Because of the different pathologists that contributed to the study, we acknowledge the nuanced nature of some of the infection types and have opted for a relative resistance (R) and susceptibility (S) score in particular with the mutants bearing in mind the confounding background genes as previously discussed.

- I have difficulties to understand the choice of the word “Resolving” in the title, ambiguous to me.

Response:

The title has now been changed. Our sincere appreciation to Dr Yue Jin for the overall feedback that has helped improve the quality of the manuscript.

Reviewer: Yue Jin

Review completed Nov 15, 2019

Reviewer #4 (Remarks to the Author):

Jianping Zhang et al. studied wheat stem rust resistance genes, Sr26 and Sr61. Both genes are originated from the wheat relative - *Thinopyrum ponticum*, which demanded a unique cloning approach when the work is done in wheat. Authors used the RenSeq approach and identified two NLR genes that were effective for stem rust resistance in wheat. Cytogenetic studies illustrated a small introgressed Ae segments with Sr26 and Sr61 in wheat. The stack of both genes in one chromosome segment enables the co-deployment of both genes in cultivated wheat. This study generated useful data for wheat community, the manuscript is well written, and conclusions are partially convincing. However, some results and conclusions require further clarification and more experimental data.

Major comments

1. “... from tall wheat grass” in the title is misleading. Readers may think both genes were directly cloned from the tall wheat grass. Reword the title to make it nonambiguous.

Response:

The title has now been changed.

2. Authors indicated the broad-spectrum wheat stem rust resistance of Sr26 and Sr61. How many Pgt races were tested for each gene? Will the tested Pgt races represent a broad-spectrum of virulence of all current Pgt races worldwide? Authors need clarify how these genes confer a broad-spectrum resistance to Pgt or delete the concept.

Response:

Lines with *Sr26* have been widely tested in rust labs for 50 years; lines with *Sr61* have not been as widely tested, but none-the-less still examined. Virulent races have never been reported for either gene so far as we are aware. This qualifies these as genes broad spectrum in our opinion.

3. Authors stated that “Whether or not the durability of Sr26 and Sr61 were due to a combined effect from multiple R genes located on the introgressed Th. ponticum alien segment remain unknown.” If this is the case, how can they be certain that the cloned R genes are Sr26 and Sr61?

Response:

The reviewer is correct on this point and we have addressed this concern extensively above in the opening comments to the editor. Rather than re-iterate these points we direct the reviewer to comments above to the editor to address this point and also refer them to the new discussion where this point is extensively addressed.

4. MutRenSeq was used to clone both genes. Apparently, authors assumed that Sr26 and Sr61 are both NLR genes. As discussed in the paper, the region may contain a cluster of resistance genes. Is it possible that the cloned genes are not the true Sr26 and Sr61 genes? They will need clarify in the manuscript what drove them to believe that Sr26 and Sr61 must be NLR genes and then pursued MutRenSeq.

Response:

Please see answer (3) as it relates to the issues raised in (4) and refer to the opening comments to the editor and new discussion. The genes we have isolated clearly account for most and possibly all of the resistance conferred by these introgressed segments.

5. Authors stated that “We identified five susceptible ethyl methanesulfonate (EMS)-induced mutants from the Sr26-carrying wheat genetic stock, Avocet+Lr46 We also identified eight susceptible EMS mutants derived from line W3757 thereby enabling isolation of Sr61.” Authors should tell the mutant population sizes of Avocet+Lr46 and W3757, numbers of susceptible mutants, and numbers of mutant lines with non-synonymous substitutions or deletions in the two cloned NLR genes.

Response:

Size of the populations has been added into the manuscript. For *Sr26*, we obtained six mutants from 1270 M₂ lines and five were used for MutRenSeq. For *Sr61*, we obtained 8 mutants from 1,837 M₂ lines and 6 were used. The number of point mutations and deletion mutants recovered is now also stipulated in the results ie. for *Sr26* 1 deletion mutant and 4 point mutations underwent RenSeq analysis. For *Sr61* 6 point mutation mutants underwent RenSeq while two deletion mutants were not included.

6. Apparently, authors believe that the two NLRs represent Sr26 and Sr61 and they stated like this in the manuscript. Is it possible that there are other resistance genes in the designated locus?

Response:

Again, we have addressed this concern extensively above in the opening comments to the editor. Rather than re-iterate these points we direct the reviewer to comments above to the editor to address this point and also refer them to the new discussion where this point is extensively addressed. We cannot rule out the possibility that additional resistances could be located on these segments (which we have covered in the revised Discussion section) but we have shown that the R genes we have cloned account for much and possibly all of the resistance specificities observed for these two introgressions.

7. “In the current study, none of the three mutations in the LRR domain of Sr61 resulted in a new resistance specificity based on the Pgt races tested.” What has been done to make this statement?

Response:

Although we showed that the three *Sr61* LRR domain mutants are all equally susceptible to the races tested (first three races listed in Table S5), we agree with the reviewer that this sample is not large enough to conclude an absence of new specificity. This is only a minor point mentioned in the manuscript and it has been removed.

8. What is the purpose of structure modelling in Figure S2b? Does this help resolve the mechanisms of the mutation? If not, they should delete modelling.

Response:

We believe this supplementary figure will be of interest to other researchers interested in structure/function modelling and mutagenesis of NLR proteins. Given it is such a modest addition in the supplemental figures section and is only briefly mentioned in the manuscript, it has been removed.

9. Authors proved that T1 plants of Fielder: Sr26: Sr22RE and Fielder: Sr26: Sr33RE were resistant to the Pgt race 98-1,2,(3),(5),6. What is the result of T1 plants of Fielder: Sr26: NativeRE? This is important data but missing in the manuscript. Readers want to know whether the native Sr26 alone (viz. 917bp + 6066bp + 263bp) is sufficient to confer an inheritable resistance.

Response:

We have added images of the rust analysis done on T₁ transgenics containing the Sr26: NativeRE to the manuscript in Figure S3 as requested.

10. Agrobacterium mediated transfer can generate single or low-copy insertions. What is the copy number of the transgene in selected T1 families? In their T1 test with Pgt race 98-1,2,(3),(5),6, what is the segregation ratio between resistance and susceptibility? Is there a correlation between copy numbers, cDNA levels and stem rust severity? The data should be provided to help readers understand how the gene works.

Response:

As requested, we have added the copy number analysis result for *Sr26* T₁ and both copy number and cDNA expression level analysis results for *Sr61* T₁. There was little difference in resistance phenotypes seen between Sr26 transgenics and the fact that multiple transgenics with an entirely native construct gave a wild type resistance phenotype, largely precludes aberrant expression as a possible cause for resistance. In contrast as only a chimeric *Sr61* construct was used (Sr61: Sr26RE) we undertook both copy number and expression level

experiments. Only two *Sr61* transgenic lines contained the transgene but were susceptible. Both lines were subsequently shown to have a very low expression level of *Sr61*. In contrast the remaining transgenics which were all resistant showed *Sr61* expression levels that were similar (but tending less than) the endogenous genes. No general correlation between expression level and transgene copy number was seen in these lines.

11. Author stated that “Thus the gene candidate was not only necessary but also sufficient to confer Sr26 resistance (Tables S1, S3).” Have they compared the resistance spectrum of transgenic plants and the Sr26 donor? If not, how do they know the candidate gene confer the Sr26 resistance?

Response:

Please see answers in questions (3) and (6) from the same reviewer.

12. Given that the additional Sr22 and Sr33 promoter and terminator sequences did not abolish Sr26 gene function, does the presence of Sr22 and Sr33 RE sequences boost the Sr26 gene function?

Response:

As stated in question (10) there was little difference in resistance phenotypes seen between Sr26 transgenics containing either 3 constructs and the fact that multiple transgenics with an entirely native construct gave a wild type resistance phenotype, largely precludes aberrant expression as a possible cause for resistance for these transgenics.

13. Unfortunately, we don't see the complementation data of the Sr61 candidate. Although the complementation does not necessarily tell the identified NLR is Sr61, it will support that the identified NLR is effective for stem rust resistance. A complementation experiment should be done and included for the Sr61 candidate.

Response:

Complementation has been undertaken and successfully confirmed for the *Sr61* gene as described above.

14. In Figure S4, the PCR products did not migrate at the same line. Could this be caused by amplifying different alleles? Did they sequence the PCR product to confirm their identity? Which Sr26+Sr61 recombinant (+) was used in S4a, 378/15, 388/15, or 376/15?

Response:

The different rate of migration was just a gel artefact. We have repeated the assay and now show a new image.

15. In Figure S6c, it appears that both W3757 behaved similar to Sr61 M2 in response to TTKTF, TTKTT, and TTRTF. If this is true, the mutation of the Sr61 candidate did not compromise the resistance to these races. Another gene should be involved in conferring resistance in the W3757 background.

Response:

We have addressed the background resistance present in W3757 throughout the new manuscript. Discovering background resistance is not at all uncommon when a large and diverse range of *Pgt* isolates are being tested on unimproved germplasm. It does of course mask the resistance gene of interest as we have noted for the *Sr61* response to 3 isolates.

16. Wheat has three type of rust problems. In the phylogenetic tree, both stem rust and leaf rust R genes are analyzed. As reported, Yr5 and YrAS2388 are also NLR genes, are these genes included in this tree? Any other yellow rust resistance genes should be included here? It is interesting to see how all wheat rust resistance genes are classified in one tree.

Response:

We have added Yr5, Yr7, and YrAS2388, and the recently cloned *YrU1* to the phylogenetic tree as requested by the reviewer.

17. To provide the source of all accessions in Figure 3a, it is useful to add species (eg. Os, Hv, and Ae) as prefix for each accession and explain the prefix in legend. In Data S1, add two columns, for species and diseases, respectively.

Response:

We tried to put the prefix into the figure, but it looked a bit too crowded. Therefore, we added all the species and disease information in Table S7.

18. This study misses the expression profiles of both genes, regarding to the developmental stages, tissue specificity, and pathogens. The expression profiles are necessary for understanding the regulation of these genes.

Response:

As we mentioned in the response to question No.10, we have examined the expression of the transgene and endogenous *Sr61* gene given that we had only used heterologous regulatory sequences for this complementation assay. We have phenotyped both *Sr26* and *Sr61* in seedlings and adult plants and shown that the resistances are expressed throughout. It is not clear what an RNA expression analysis will contribute as the resistance phenotype during plant development we believe is more insightful. In addition, the majority of NLR genes examined tend to show constitutive low-level expression.

19. Because the two NLR genes were effective, they might discuss how to manage gene stewardship to increase resistance durability. They might also discuss the fate of the two NLR genes after they are deployed in large area in regions with severe stem rust epidemics.

Response:

These are interesting points but given the word restrictions of this manuscript we unfortunately didn't have much space to address these points in detail previously. We do point out that the combined deployment of both genes by combining them on a nonrecombinogenic segment is great way to improve the gene stewardship of both and is likely to increase the durability of both, which is the main message of the paper. It is of course difficult to predict the fate of resistance genes as some single genes can occasionally remain remarkably durable while many do not.

Minor comments

1. Some affiliations miss postcodes.

Response:

Revised accordingly.

2. In the introduction, they stated that “Sr26 was transferred to wheat chromosome 6A ... and this resistance has remained effective against all known Pgt races.... to date no known virulence has been reported for Sr61.” This again is related to the wide-spectrum term. It will be helpful to introduce how the two genes were deployed in production, such as how many years on how much hectares.

Response:

We include the following sentences in the manuscript which covers these points. “*Sr26* has been deployed in a number of Australian wheat cultivars since 1971 and has likely fulfilled the definition of durable resistance. *Sr61* on the other hand has not been deployed in a cultivar.”

3. Add a period sign after “... deletion of a linked marker (Figure1a; Table S1)”

Response:

Revised accordingly.

4. Italicize Lr46 in “Avocet+Lr46”.

Response:

We feel in the NILs/line names, the gene should not be italicized.

5. Add space in “Figure1b” in “LRR motifs at the C-terminus (CNL) (Figure1b)”

Response:

We re-designed Figure 1b.

6. Delete “translated” from “and effects in the predicted translated Sr26 and Sr61 proteins” in the Figure 1 legend.

Response:

We re-designed Figure 1b.

7. In Figure 1, use a simple straight line to replace the while dotted blocks (introns). The NB-ARC motif should exclude the intron region.

Response:

We re-designed Figure 1b.

8. In Figure 1b, 499S was incorrectly placed.

Response:

We re-checked the position and it should be correct.

9. Should it be “S1b” in “were analysed by NLR gene capture and sequencing (Figures 1a, S1a).”

Response:

Revised accordingly.

10. “Sr61 M1” is missing in Figure S2a.

Response:

Revised accordingly.

11. Authors stated that “A single contig of 3,519 bp was identified that contained nucleotide changes in five of the mutants (Figure S1b) identified an additional alteration in the sixth mutant (M6).” Supply IDs for the first five mutants and delete “additional”.

Response:

Revised accordingly.

12. In Figure S2a, a note is required for the blue frame in the Sr26 sequence prior to the residue 486 requires.

Response:

Revised accordingly.

13. In Figure 2a. “Sr26 ORF” should be changed to “Sr26 ORF-corresponding genomic region”

Response:

We re-designed the figure.

14. In Figure 2b, add labels to the picture directly, instead of using 1 to 7.

Response:

Revised accordingly.

15. In Figure S3a, missing susceptible control Fielder. “(A) .. (B)...” should be changed to small case.

Response:

Revised accordingly.

16. In Table S1, progeny test (M3) must be done for the S sib, but missing this specific information. This manuscript used M1 to M8 as IDs for Sr61, which are confusing with the mutant generation numbers. To specify a generation, it is better use M with numbers as subscripts. A note of SrB = Sr61 should be added, or replacing SrB with Sr61. Clarify if the R sib (M2) and Mutant (M2) were derived from the same M1 plant.

Response:

Revised accordingly. We changed “M1-M8” into “Mutant1-Mutant8” to avoid the confusion with the generation numbers. We changed *SrB* to *Sr61* in this table. The M2 R sib and the mutant listed in the table were derived from the same M1 plant.

17. In Figure S7a-b, d, explain how to understand the disease severity scores, such as 20MRR; in Figure S7c, revise “Sr26mutant 12S1”

Response:

Revised accordingly.

18. Use “3b” in “Sr13 (58.46% identity at protein level) (Figure 3B)”; delete “, 3b” in “barley Mla R gene family (Figures 3a, Clade I, 3b).”

Response:

Revised accordingly.

19. In Table S2, add notes or reference to clarify infection type scores; explain how different scores for the same event were collected, eg. 1+2-2; provide infection types for T1 generations.

Response:

Revised accordingly, reference was added.

20. In Table S3, specify “Avocet” used.

Response:

Revised accordingly.

REVIEWERS' COMMENTS

Reviewer #1 (Remarks to the Author):

NCOMMS-19-35922A

Thanks to the authors for your detailed reply to the reviews and your revisions. I am happy with the reply and revisions made to the manuscript. I have two minor comments for your consideration, but beyond that recommend that the manuscript be accepted.

The first line of the abstract doesn't read well. Consider moving "in Europe and Africa" to after (Pgt).

Pg3 – I thought that in general Sr should not be italics when referring to a gene class but it should when it is part of a gene name. Consider changing the font style for Sr when it is not part of a gene name.

Reviewer #2 (Remarks to the Author):

In the revised version, authors addressed all my concerns and I don't have additional comment.

Reviewer #3 (Remarks to the Author):

Dear authors/editors:

The revised manuscript have addressed all my concerns. I am particularly appreciative of the addition of large amounts critically important new data, especially the complementation experiment on Sr61, origins of the fungal isolates, and photographic documentation of the infection responses at various stages.

Yue Jin

Reviewer #4 (Remarks to the Author):

The cloning of Sr26 and Sr61 are important for stem rust resistance breeding and for understanding R genes. Authors have made significant improvement of this manuscript. It will be nice if authors consider the following comments.

1. Abstract should be rewritten. There is too much background information, not informative for what is gained in the current study.
2. Some long sentences should be simplified, such as "The re-emergence in Europe and Africa of wheat stem rust caused by *Puccinia graminis* f. sp. *tritici* (Pgt) reinforces the ongoing need for durable stem rust (Sr) resistance gene deployment in wheat." and "These genes were targeted due to their effectiveness against Ug99 and other Pgt races and their sequences now provide perfect markers and diagnostic tools for marker assisted breeding and opportunities for potential transgene deployment."
3. Some places need references or original reference.
4. Delete "broad spectrum" in two places of your main text.

5. "The assembled genomic sequence available for the Sr26 candidate included 917 bp 5' of the translational start codon and 263 bp of 3' of the stop codon (DELETE "OF" AFTER 263 bp; ADD Fig. 2a)".

6. The other two, designated Sr26:Sr22RE and Sr26:Sr33RE, combined the Sr26:NativeRE construct with 5' and 3' regulatory elements derived from wheat stem rust resistance genes Sr22 and Sr33, respectively (ADD Fig. 2a).

7. Fig 3f: Why did W3757 show clear pustules? All transgenic T1 plants were lower in expression than W3757, indicating the native regulatory sequences are incomplete.

8. To ensure the expression of the candidate gene a heterologous construct designated Sr61:Sr26RE was produced encoding the Sr61 genomic sequence and the 5' and 3' Sr26 regulatory elements shown above to function in the Sr26:NativeRE construct (ADD Fig. 2a).

9. Remarkably, six stem rust resistance genes (Sr24, Sr25, Sr26, Sr43, Sr44, Sr61), four leaf rust resistance genes (Lr19, Lr24, Lr29, Lr38), two powdery mildew resistance genes (Pm40, Pm43), two barley yellow dwarf virus resistance genes (Bdv2, Bdv3), and a recently cloned wheat Fusarium head blight resistance gene (Fhb7) were transferred from *Th. spp.* to wheat26 (add REVIEWED in 26).

10. In contrast, Yr10, YrU1 and YrAS238833 encode canonical NLR proteins (Delete Yr10 since its identity is controversial).

Response to the reviewers' comments

REVIEWERS' COMMENTS

Reviewer #1 (Remarks to the Author):

NCOMMS-19-35922A

Thanks to the authors for your detailed reply to the reviews and your revisions. I am happy with the reply and revisions made to the manuscript. I have two minor comments for your consideration, but beyond that recommend that the manuscript be accepted.

The first line of the abstract doesn't read well. Consider moving "in Europe and Africa" to after (Pgt).
Revised and highlighted as shown in the abstract.

Pg3 – I thought that in general Sr should not be italics when referring to a gene class but it should when it is part of a gene name. Consider changing the font style for Sr when it is not part of a gene name.

Revised and highlighted (pages 3, 4, 11, 12).

Reviewer #2 (Remarks to the Author):

In the revised version, authors addressed all my concerns and I don't have additional comment.

Reviewer #3 (Remarks to the Author):

Dear authors/editors:

The revised manuscript have addressed all my concerns. I am particularly appreciative of the addition of large amounts critically important new data, especially the complementation experiment on Sr61, origins of the fungal isolates, and photographic documentation of the infection responses at various stages.

Yue Jin

Reviewer #4 (Remarks to the Author):

The cloning of Sr26 and Sr61 are important for stem rust resistance breeding and for understanding R genes. Authors have made significant improvement of this manuscript. It will be nice if authors consider the following comments.

1. Abstract should be rewritten. There is too much background information, not informative for what is gained in the current study.

Revised and highlighted. The revised abstract has only one introductory sentence and the rest devoted to the findings of the current study.

2. Some long sentences should be simplified, such as “The re-emergence in Europe and Africa of wheat stem rust caused by *Puccinia graminis* f. sp. *tritici* (Pgt) reinforces the ongoing need for durable stem rust (Sr) resistance gene deployment in wheat.” and “These genes were targeted due to their effectiveness against Ug99 and other Pgt races and their sequences now provide perfect markers and diagnostic tools for marker assisted breeding and opportunities for potential transgene deployment.”

Revised and highlighted (page 3).

3. Some places need references or original reference.

Revised and highlighted in the manuscript. We added the new references in the Discussion section, paragraph 4 as new references No. 33, 35, 36, 37, and 38.

4. Delete “broad spectrum” in two places of your main text.

Revised accordingly.

5. “The assembled genomic sequence available for the Sr26 candidate included 917 bp 5’ of the translational start codon and 263 bp of 3’ of the stop codon (DELETE “OF” AFTER 263 bp; ADD Fig. 2a)”.

Revised and highlighted in the manuscript (page 5).

6. The other two, designated Sr26:Sr22RE and Sr26:Sr33RE, combined the Sr26:NativeRE construct with 5’ and 3’ regulatory elements derived from wheat stem rust resistance genes Sr22 and Sr33, respectively (ADD Fig. 2a).

Revised and highlighted in the manuscript (page 6).

7. Fig 3f: Why did W3757 show clear pustules? All transgenic T1 plants were lower in expression than W3757, indicating the native regulatory sequences are incomplete.

The wheat line W3757, as described and discussed in the manuscript carried additional resistance genes to *Sr61*. In Fig 3f, the *Pgt* race used appears to have knocked out some of the background genes. The reviewer is correct on saying the incomplete native regulatory sequence is one possible reason for causing this. In addition, it is common that the transgenics may show slightly different phenotypes in comparing with the wildtype due to different genetic background between Fielder and W3757.

8. To ensure the expression of the candidate gene a heterologous construct designated Sr61:Sr26RE was produced encoding the Sr61 genomic sequence and the 5’ and 3’ Sr26 regulatory elements shown above to function in the Sr26:NativeRE construct (ADD Fig. 2a).

Revised and highlighted in the manuscript (page 7).

9. Remarkably, six stem rust resistance genes (Sr24, Sr25, Sr26, Sr43, Sr44, Sr61), four leaf rust resistance genes (Lr19, Lr24, Lr29, Lr38), two powdery mildew resistance genes (Pm40, Pm43), two barley yellow dwarf virus resistance genes (Bdv2, Bdv3), and a recently cloned wheat *Fusarium* head blight resistance gene (Fhb7) were transferred from *Th. spp.* to wheat26 (add REVIEWED in 26).

Revised and highlighted in the manuscript (page 10).

10. In contrast, Yr10, YrU1 and YrAS238833 encode canonical NLR proteins (Delete Yr10 since its identity is controversial).

Yr10 has been deleted accordingly.